

# Efficiently handling constraints in mixed-integer nonlinear programming problems using gradient-based repair differential evolution

Daniel Molina-Pérez[1], Edgar Alfredo Portilla-Flores[2], Efrén Mezura-Montes[3], Eduardo Vega-Alvarado[1] and María Bárbara Calva-Yañez[2]

[1] Centro de Innovación y Desarrollo Tecnológico en Cómputo, Instituto Politécnico Nacional, Ciudad de Mexico, México
[2] Unidad Profesional Interdisciplinaria en Ingeniería Campus Tlaxcala, Instituto Politécnico Nacional, Tlaxcala, México
[3] Artificial Intelligence Research Institute, University of Veracruz, Veracruz, México

## ABSTRACT

Mixed integer nonlinear programming (MINLP) addresses optimization problems that involve continuous and discrete/integer decision variables, as well as nonlinear functions. These problems often exhibit multiple discontinuous feasible parts due to the presence of integer variables. Discontinuous feasible parts can be analyzed as subproblems, some of which may be highly constrained. This significantly impacts the performance of evolutionary algorithms (EAs), whose operators are generally insensitive to constraints, leading to the generation of numerous infeasible solutions. In this article, a variant of the differential evolution algorithm (DE) with a gradient-based repair method for MINLP problems (G-DEmi) is proposed. The aim of the repair method is to fix promising infeasible solutions in different subproblems using the gradient information of the constraint set. Extensive experiments were conducted to evaluate the performance of G-DEmi on a set of MINLP benchmark problems and a real-world case. The results demonstrated that G-DEmi outperformed several state-of-the-art algorithms. Notably, G-DEmi did not require novel improvement strategies in the variation operators to promote diversity; instead, an effective exploration within each subproblem is under consideration. Furthermore, the gradient-based repair method was successfully extended to other DE variants, emphasizing its capacity in a more general context.

# INTRODUCTION

Mixed integer nonlinear programming (MINLP) problems represent the subset of optimization problems that involve continuous and discrete/integer decision variables, as well as nonlinear functions. In engineering and related fields, discrete and integer variables are commonly used to represent values restricted to specific sets. Discrete variables are often used when values are limited to standard elements, such as rod diameters, structural

Corresponding authors
Daniel Molina-Pérez,
dmolinap1800@alumno.ipn.mx
Edgar Alfredo Portilla-Flores,
aportilla@ipn.mx

beam dimensions, and commercial resistance values. Integer variables are frequently employed to represent groups of identical elements, such as the number of products manufactured, pins in an electrical connector, or gear teeth in a gear train.

A MINLP problem can be expressed by Eq. (1):

$$
\begin{aligned}
& \min f(\mathbf{x}, \mathbf{y}) \\
\text{subject to}: \quad & g_i(\mathbf{x}, \mathbf{y}) \leq 0, \ i = 1, \ldots, n \\
& h_j(\mathbf{x}, \mathbf{y}) = 0, \ j = 1, \ldots, m \\
& x_k^l \leq x_k \leq x_k^u, \ k = 1, \ldots, n_1 \\
& y_q^l \leq y_q \leq y_q^u : \text{integer}, \ q = 1, \ldots, m_1 \\
& [\mathbf{x}, \mathbf{y}] \in \Omega
\end{aligned}
\tag{1}
$$

where $f(\mathbf{x}, \mathbf{y})$ is the objective function (OF) of the problem, $\mathbf{x}$ is a vector of continuous variables, $\mathbf{y}$ is a vector of integer variables. $x_k^l$ and $x_k^u$ are the lower and upper bounds of $x_k$, respectively, whereas $y_q^l$ and $y_q^u$ are the lower and upper bounds of $y_q$, respectively. The decision variable space is defined by $\Omega$, and $g_i(\mathbf{x}, \mathbf{y})$ and $h_j(\mathbf{x}, \mathbf{y})$ denote the $i$th inequality constraint and the $j$th equality constraint, respectively.

One key challenge in MINLP problems is that the presence of integer variables generates a nonconvex search space, which can lead to a combinatorial explosion in the number of possible solutions. This is because the discrete variables create a set of discontinuous feasible parts, and the optimization algorithm must explore each part separately. Figure 1 illustrates a generic MINLP problem with a search space defined by two variables, $x$ and $y$. The variable $x$ can take all real values, and $y$ is limited to integer values $\{0, 1, \ldots, 4\}$. The gray area is the feasible region defined by the constraints. Contour lines connect points from discontinuous parts with equal fitness values through steps. However, these steps solely provide a visual indication of the function's behavior in the search space, which cannot be evaluated for non-integer values of $y$. The figure reveals another important aspect of MINLP problems: the high variability in the sizes of the discontinuous feasible parts (red lines). This phenomenon is attributed to the coexistence of constraints and integer variables.

Stochastic optimization methods such as EAs have been updated to address the challenges posed by MINLP problems, *e.g.*, genetic algorithms (*Deep et al., 2009*), evolution strategies (*Costa & Oliveira, 2001*), differential evolution (*Lampinen & Zelinka, 1999*), particle swarm optimization (PSO) (*Wang, Zhang & Zhou, 2021*), ant colony optimization (*Liao et al., 2013*), among others (see *Boukouvala, Misener & Floudas, 2016* and *Ploskas & Sahinidis, 2022* for a comprehensive survey of algorithms and software). EAs have successfully been applied to solve remarkable MINLP problems, such as the design of an interplanetary space mission trajectory (based on the Galileo mission from NASA in 1989) (*Schlüter et al., 2013*), optimal control problems for an F8 aircraft (*Sager, 2011*), as well as optimal batch plant design problems (*Ponsich & Coello, 2011*; *Ji & Gu, 2024*). Recently, EAs have been used to optimize hyperparameters for convolutional neural networks (*Liu & Wang, 2023*), reduce neural network structure complexity (*Sildir, Sarrafi & Aydin, 2022*), and design a multimodal hub-and-spoke transportation network for emergency

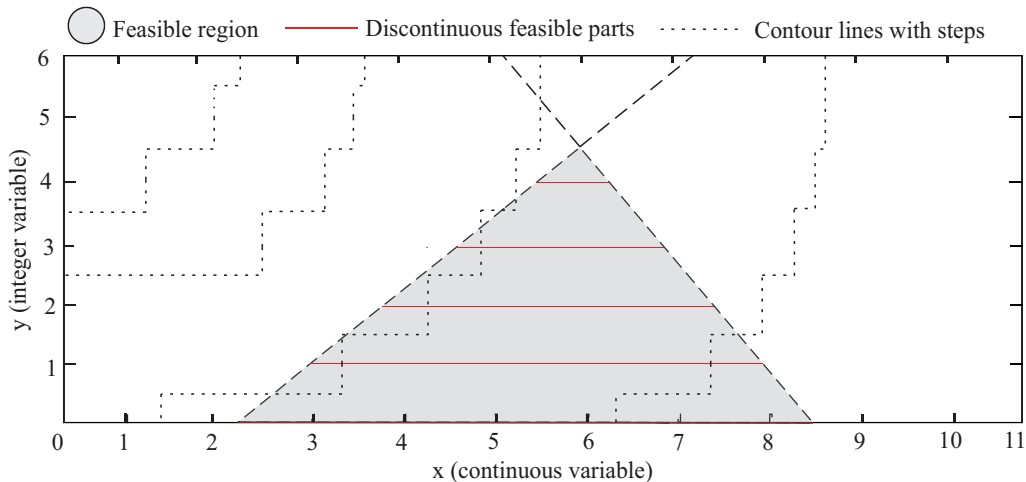

**Figure 1 Hypothetical example: The gradient area represents the feasible region defined by the constraints, and the red lines are the discontinuous feasible parts that also satisfy the integer constraints.**

relief during the COVID-19 pandemic (*Li et al., 2023*). However, premature convergence remains a major concern for EAs when addressing MINLP problems with variable-size discontinuous feasible parts. In such cases, the population tends to converge on the biggest feasible parts because the EAs tend to detect feasible solutions more easily within these regions. Consequently, this approach may limit the exploration of smaller feasible parts that could contain better solutions.

As can be seen in Fig. 1, a MINLP problem can be expressed as a set of **k** subproblems, where **k** corresponds to all the admissible values of $y$, in this case, $\{0, 1, .., 4\}$. Note that the global minimum is the optimal solution for all subproblems. However, due to the dimension of **k** in several problems, its overall combinatorial enumeration could be computationally expensive or even prohibitive. Two important issues are derived from this approach: (i) an efficient combinatorial exploration in the space of discrete variables, and (ii) an efficient exploration and exploitation within each generated subproblem.

Commonly used operators in EAs are often blind to constraints, making it difficult to solve highly constrained problems (*Craenen, Eiben & Marchiori, 2001*; *Hamza, Essam & Sarker, 2015*). This has a high cost for algorithm performance in solving highly constrained MINLP problems due to the challenge of generating feasible solutions. To address this point, the operators in diverse proposals have been modified to achieve higher diversity and more opportunities to explore each subproblem. As optimization problems become more complex, particularly within the realm of MINLP problems, the challenge of effectively exploring highly constrained spaces becomes a substantial obstacle. This work proposes a DE variant with a gradient-based repair method for MINLP problems (G-DEmi). Unlike previous methods focused on diversification to enhance performance, G-DEmi takes a distinctive approach. It attempts to improve promising but infeasible solutions in different subproblems by using gradient information extracted from the constraint set. The proposal was evaluated using benchmark problems selected from

*Liu et al. (2021a)*, *Datta & Figueira (2013)*, *Liu et al. (2023)*, and *MINLPLib (2023)*, which encompassed several challenges posed by mixed variables. Additionally, the subway optimization problem was also addressed as a real-world problem. The proposal was rigorously compared with other state-of-the-art EAs using descriptive and inferential statistics. The results show that the proposal consistently outperforms competitors in terms of solution quality and algorithm robustness. Notably, G-DEmi achieved high success rates even in highly constrained problems where other algorithms failed to find feasible solutions. These findings indicate that the gradient-based repair method significantly contributes to the performance of G-DEmi across a wide range of MINLP problems. This algorithm proved its efficiency in exploring each generated subproblem since it did not require new operators to promote a higher diversity but the conventional DE operators. Furthermore, the repair method was successfully extended to other DE variants, highlighting its robustness and effectiveness in a more general context.

This article is organized as follows: "Related Work on Evolutionary Algorithms for MINLP Problems" presents an overview of the state-of-the-art works referred to the development of EAs for MINLP problems. "Base Methods" gives a brief description of DE and the gradient-based repair method. The proposed algorithm, G-DEmi, is introduced in "Proposed Algorithm". "Experimental Study" describes the experimental study conducted on G-DEmi, and "Real-World Case Study" addresses the real-world case study. "Conclusions" presents the final considerations.

## RELATED WORK ON EVOLUTIONARY ALGORITHMS FOR MINLP PROBLEMS

In the field of MINLP, several techniques have been employed to address its complexities. The initial approach involved genetic algorithms with binary coding to represent integer and real variables (*Li & Gen, 1996*). This approach has been extended to other algorithms such as particle swarm optimization (*Datta & Figueira, 2011*), differential evolution (*Datta & Figueira, 2013*), and artificial bee colony (*Wang et al., 2017*). However, the effectiveness of these techniques is significantly affected by problem dimensionality and the ranges of the integer and real variables.

To address the issues of integer variables, some studies proposed truncation or rounding methods during the optimization process (*Deep et al., 2009*; *MathWorks, 2023*). However, these operations may introduce search bias or drastic reductions in diversity, potentially impacting the performance of the algorithm. More refined strategies have been proposed instead of simple rounding or truncation. A triangular mutation rule was introduced by *Mohamed (2017)*, which is based on three randomly selected vectors and the difference vectors between the best, medium, and worst individuals within that triplet. This operator was designed to improve the global and local search capabilities and to increase the convergence speed of DE. *Jalota & Thakur (2018)* proposed a variant of GA with BEX crossover and power mutation. Additionally, a truncation technique based on the floor and ceiling function was applied to maintain the randomness in the evolutionary process.

Another variation of DE with cluster strategy into the mutation operator was presented by *Mohamed et al. (2019)*. This approach selects three random elements from different clusters to perform the mutation. *Song et al. (2023)* proposed a hybrid sine and cosine algorithm that uses a position update formula to enhance its search capability. In addition, they proposed three mutation strategies with different balances between exploration and exploitation to prevent local convergence.

Another approach involves estimation-of-distribution algorithms (EDAs) combined with different strategies for solving MINLP problems. *Wang et al. (2019)* proposed an EDA with an adaptive-width histogram model and a learning-based histogram model to handle continuous and discrete variables, respectively. *Sun & Gao (2019)* proposed a probabilistic model to update the discrete variables in PSO. It uses social cognitive and self-cognitive coefficients to generate a probability distribution. Subsequently, *Wang, Zhang & Zhou (2021)* employed a learning-based histogram model for discrete variables in PSO. *Li et al. (2021)* proposed an EDA algorithm for mixed variables to find the best parameters of a convolutional neural network. *Peng et al. (2021)* implemented a histogram model in DE for handling discrete variables.

Other studies have presented specific strategies for directly addressing discrete variables. *Schlüter (2012)* proposed a discretized version of the multi-kernel function for the mixed-integer distributed ant colony optimization (MIDACO) algorithm. MIDACO employs an oracle penalty function proposed by *Schlüter & Gerdts (2010)*, which significantly biases the search towards promising regions of the solution space. *Lin et al. (2018)* proposed a set-based DE approach that preserves the original search mechanism, avoiding space transformation. *Liu, Li & Ge (2022)* proposed a hybrid quantum annealing-double-elite spiral search algorithm to address MILNP problems. The algorithm was applied to solve the integer subproblems using a quantum-tunneling-based annealing mechanism.

Recent works have focused on overcoming the limitations due to search spaces in MINLP problems. *Liu et al. (2021b)* proposed a multi-objective DE aimed at identifying solutions that equally satisfy integer and fitness conditions. In *Liu et al. (2021a)*, a cutting and repulsion strategy was proposed to suppress unpromising discontinuous feasible parts during exploration and to escape from local optima. In *Liu et al. (2023)*, a surrogate-assisted DE was proposed, featuring an adaptive pre-screening operation to prevent excessive exploitation in local regions, which restricts the number of individuals sharing the same integer values. *Hamano et al. (2022)* suggested a strategy to generate integer variables with a lower-bounding marginal probability model to avoid stagnation caused by the discretization granularity in the search space. In *Molina-Pérez et al. (2022, 2023)*, an EDA was presented, where a link strategy between the histogram model and the $\varepsilon$-constrained method reduces the influence of the larger discontinuous feasible parts in the exploration. A kriging-assisted DE was proposed to deal with mixed-integer variables by *Liu et al. (2024)*. Promising regions near the feasible region are created to promote exploration from infeasible regions. Solutions are then guided to promising feasible regions through local search. *Molina-Pérez et al. (2024)* proposed a DE variant that explores

promising regions from infeasible contours by considering a group of "good fitness-infeasible solutions". This approach aims to reduce the vulnerability of solutions to being attracted by non-promising large discontinuous feasible parts.

While numerous proposals exist to address MINLP problems, certain studies have focused on particular problems or relatively simple problem sets. In several cases, the implementation of additional operators has been required to prevent premature convergence in the search space, although the underlying reasons for this have not been explicitly stated. Only in recent works the distinctive landscape that mixed variables create within the search space has been clearly addressed. This perspective, involving disconnected feasible regions of varying sizes, has enabled more efficient strategies for solving increasingly complex problems.

In this context, gradient-based repair methods have shown high effectiveness in overcoming the limitations of EAs in handling constraints. The first and most commonly used technique was proposed by *Chootinan & Chen (2006)*. It uses Newton's root-finding method to repair infeasible solutions. More recently, several gradient-based methods have been introduced, including repair method on surrogates of the constraint functions (*Koch et al., 2015*), random direction repair method (*Xu et al., 2021*), heuristic repair with historical information (*Du et al., 2022*), and adaptive repair method (*Yang et al., 2023*). However, these tools haven't received proper attention in MINLP problems, despite their potential utility, especially considering the search space comprising multiple constrained subproblems. We propose G-DEmi, with a gradient-based repair method to rectify promising infeasible solutions in different subproblems using gradient information derived from the constraint set. Unlike prior methods, this novel approach aims to directly address the constraint challenges by reducing the insensitivity of DE to constraints in MINLP problems. Notably, G-DEmi improves the performance without needing new operators for higher diversity than those employed in the conventional DE.

## BASE METHODS

### Differential evolution

*Storn & Price (1997)* proposed a population-based optimization algorithm, DE, that has gained popularity for its high success rate and straightforward implementation. DE consists of four fundamental processes: initialization, mutation, crossover, and selection, which are repeated in each cycle.

The initial population $P$ is created through random sampling from the search space. Each individual in the population, denoted as $\mathbf{x}_i = (x_{i,1}, \ldots, x_{i,j})$, represents a solution to the problem with $x_i$ being the $i$th individual, $i = 1, \ldots, \text{NP}$. $j$ is the number of decision variables and NP is the population size. A mutant vector $\mathbf{v}_i = (v_{i,1}, \ldots, v_{i,j})$ is generated for every individual $\mathbf{x}_i$ as the base of the evolution. Several mutation operators have been proposed (*Abbasa, Ahmadb & Jabeenc, 2017*). In this study, the following versions were employed:

- DE/rand/1:

$$\mathbf{v}_i = \mathbf{x}_{r1} + F(\mathbf{x}_{r2} - \mathbf{x}_{r3}) \tag{2}$$

- DE/current-to-rand/1:

$$\mathbf{v}_i = \mathbf{x}_i + \mathrm{rand}(\mathbf{x}_{r1} - \mathbf{x}_i) + F(\mathbf{x}_{r2} - \mathbf{x}_{r3}) \tag{3}$$

- DE/best/1:

$$\mathbf{v}_i = \mathbf{x}_{best} + F(\mathbf{x}_{r1} - \mathbf{x}_{r2}) \tag{4}$$

- DE/rand-to-best/1:

$$\mathbf{v}_i = \mathbf{x}_{r1} + \mathrm{rand}(\mathbf{x}_{best} - \mathbf{x}_{r2}) + F(\mathbf{x}_{r3} - \mathbf{x}_{r4}) \tag{5}$$

where $\mathbf{x}_{best}$ is the best individual in the population; vectors $\mathbf{x}_r$ are different between each other and from $\mathbf{x}_i$, and randomly chosen from the population; and $F$ is the scaling factor.

DE executes a crossover operation between the target vector $\mathbf{x}_i$ and its mutant vector $\mathbf{v}_i$ to generate the trial vector $\mathbf{u}_i = (u_{i,1}, \ldots, u_{i,j})$. The binomial crossover is defined in Eq. (6), where the control crossover parameter CR is a value in $[0, 1]$. A uniformly distributed random number $\mathrm{rand}_j$ is generated for each decision variable $j$ in the interval $[0, 1]$, and $j_{\mathrm{rand}}$ is a random integer in the range $[1, \mathrm{NP}]$.

$$u_{i,j} = \begin{cases} v_{i,j}, & \text{if } \mathrm{rand}_j \leq \mathrm{CR} \text{ or } j = j_{\mathrm{rand}.} \\ x_{i,j}, & \text{otherwise.} \end{cases} \tag{6}$$

During the selection step, the best individual is chosen between $\mathbf{x}_i$ and $\mathbf{u}_i$ to be carried forward to the next generation. A set of feasibility rules (*Deb, 2000*) is employed to compare them as follows:

1. Between two infeasible solutions, the one with lower constraint violation is preferred.
2. If one solution is infeasible and the other one is feasible, the feasible solution is preferred.
3. Between two feasible solutions, the one with better objective function value is preferred.

## Gradient-based repair method

The technique of gradient-based repair was first introduced for genetic algorithms (*Chootinan & Chen, 2006*). *Takahama & Sakai (2009)* incorporated this approach in a modified proposal version of the constraint handler $\varepsilon$-constrained. This method uses the gradient information of the constraint set to guide the infeasible solutions toward feasible regions. The gradient can be obtained directly from the constraints or, if the analytical derivation is challenging, it can be estimated by numerical techniques such as the finite differences (*Chapra & Canale, 2010*).

$\mathbf{V}(\mathbf{x})$ is defined by Eq. (7) as a vector that contains the violation degree for each inequality ($g$) and equality ($h$) constraint in a given problem, for a particular solution vector $\mathbf{x}$. Parameters $n$ and $m$ are the number of inequality and equality constraints, respectively, and $\varepsilon$ specifies the tolerance for equality constraints. Preserving the sign of the equality violation is a crucial aspect of the method and is represented by the function $\mathrm{sgn}(h)$.

$$\mathbf{V}(\mathbf{x}) = \begin{bmatrix} \max(g_1(\mathbf{x}), 0) \\ \vdots \\ \max(g_n(\mathbf{x}), 0) \\ sgn(h_1) \cdot \max(|h_1(\mathbf{x})| - \varepsilon, 0) \\ \vdots \\ sgn(h_m) \cdot \max(|h_m(\mathbf{x})| - \varepsilon, 0) \end{bmatrix}. \tag{7}$$

The gradient matrix of these constraints with respect to the $N$ components of $\mathbf{x}$, denoted as $\nabla\mathbf{V}(\mathbf{x})$, is defined by Eq. (8):

$$\nabla\mathbf{V}(\mathbf{x}) = \begin{bmatrix} \frac{\partial g_1(\mathbf{x})}{\partial x_1} & \frac{\partial g_1(\mathbf{x})}{\partial x_2} & \cdots & \frac{\partial g_1(\mathbf{x})}{\partial x_N} \\ \vdots & \vdots & & \vdots \\ \frac{\partial g_n(\mathbf{x})}{\partial x_1} & \frac{\partial g_n(\mathbf{x})}{\partial x_2} & \cdots & \frac{\partial g_n(\mathbf{x})}{\partial x_N} \\ \frac{\partial h_1(\mathbf{x})}{\partial x_1} & \frac{\partial h_1(\mathbf{x})}{\partial x_2} & \cdots & \frac{\partial h_1(\mathbf{x})}{\partial x_N} \\ \vdots & \vdots & & \vdots \\ \frac{\partial h_m(\mathbf{x})}{\partial x_1} & \frac{\partial h_m(\mathbf{x})}{\partial x_2} & \cdots & \frac{\partial h_m(\mathbf{x})}{\partial x_N} \end{bmatrix}. \tag{8}$$

The forward approximation of finite differences provides an estimate of these derivatives, as defined in Eq. (9), where $\Delta x$ represents the step size and $\mathbf{e}_j$ is a unitary vector of the same dimension as $\mathbf{x}$, with a value of 1 for the $j$th component and 0 for the rest.

$$\nabla\mathbf{V}(\mathbf{x})_{i,j} \approx \frac{f_i(\mathbf{x} + \Delta x \cdot \mathbf{e}_j) - f_i(\mathbf{x})}{\Delta x}. \tag{9}$$

This repair method aims to transform $\mathbf{x}$ into a feasible solution, which involves setting the elements of the vector $\mathbf{V}(\mathbf{x})$ to zero. By employing Newton-Raphson's method (*Burden, Faires & Burden, 2015*), a repaired vector $\mathbf{x}_{k+1}$ can be iteratively obtained through Eq. (10), which represents a linear approximation of $\mathbf{V}(\mathbf{x}_k)$ in the direction of the origin. However, it is common that the number of variables differs from the number of constraints. In this case, the $\nabla\mathbf{V}(\mathbf{x}_k)$ matrix is non-invertible and the Moore-Penrose pseudoinverse (*Campbell & Meyer, 2009*) must be used, as shown in Eq. (11):

$$\mathbf{x}_{k+1} = \mathbf{x}_k - \nabla\mathbf{V}(\mathbf{x}_k)^{-1}\mathbf{V}(\mathbf{x}_k) \tag{10}$$

$$\mathbf{x}_{k+1} = \mathbf{x}_k - \nabla\mathbf{V}(\mathbf{x}_k)^{+}\mathbf{V}(\mathbf{x}_k) \tag{11}$$

where $\mathbf{x}_{k+1}$ and $\mathbf{x}_k$ represent the updated and current values of the vector $\mathbf{x}$, respectively, $\nabla\mathbf{V}(\mathbf{x}_k)^{-1}$ and $\nabla\mathbf{V}(\mathbf{x}_k)^{+}$ represents the inverse and the Moore-Penrose pseudoinverse matrix of the gradient matrix $\nabla\mathbf{V}(\mathbf{x}_k)$, respectively. A computationally efficient way of finding $\nabla\mathbf{V}(\mathbf{x}_k)^{+}$ is by employing singular value decomposition (*Barata & Hussein, 2012*). Before proceeding with the pseudoinverse, removing all zero elements of $\mathbf{V}(\mathbf{x}_k)$ and their corresponding values in $\nabla\mathbf{V}(\mathbf{x}_k)$ is important.

---

**Algorithm 1**    **Gradient-based repair method.**

**Input:** $\mathbf{u}, k_{max}, T_{min}$;

**Output:** $\mathbf{u}$;

1: initialize $k = 1$;

2: **while** no stopping criteria is satisfied **do**

3:    calculate $\mathbf{V}(\mathbf{x}_k^u, \mathbf{y}^u)$ in Eq. (12) according to Eq. (7);

4:    calculate $\nabla\mathbf{V}(\mathbf{x}_k^u, \mathbf{y}^u)$ in Eq. (13) according to Eq. (8);

5:     remove all zero elements of $\mathbf{V}(\mathbf{x}_k^u, \mathbf{y}^u)$ and their corresponding values in $\nabla\mathbf{V}(\mathbf{x}_k^u, \mathbf{y}^u)$;

6:     calculate pseudoinverse $\nabla\mathbf{V}(\mathbf{x}_k^u, \mathbf{y}^u)^+$;

7:     calculate $\mathbf{x}_{k+1}^u$ according to Eq. (11);

8: $\mathbf{x}_k^u \leftarrow \mathbf{x}_{k+1}^u$;

9: $\mathbf{u} = [\mathbf{x}_k^u, \mathbf{y}^u]^{\mathrm{T}}$;

10: $k \leftarrow k + 1$;

11: **end while**

**Stopping criteria:**
- $k \geq k_{max}$: maximum number of iterations reached.
- $\mathbf{V}(\mathbf{x}_k^u, \mathbf{y}^u) = 0$: all elements of the constraint violation degree vector are zero.
- $T_u \leq T_{min}$: maximum absolute difference between $\mathbf{x}_{k+1}^u$ and $\mathbf{x}_k^u$ is equal or lower than $T_{min}$.

---

# PROPOSED ALGORITHM

## Gradient-based repair method for MINLP problems

As mentioned above, MINLP problems can be analyzed as multiple subproblems, some of which may be highly constrained. This impacts the performance of EAs, whose operators are typically insensitive to constraints, leading to the generation of many infeasible solutions. We propose a basic DE implementation with a gradient-based repair method for MINLP problems, G-DEmi. The repair strategy explores the subproblems independently to improve the exploration inside them. Specifically, for repairing a vector with mixed variables $[\mathbf{x}, \mathbf{y}]$, only the real variables $\mathbf{x}$ are modified while the integer variables $\mathbf{y}$ remain fixed.

In contrast to the variants previously described (*Chootinan & Chen, 2006*; *Takahama & Sakai, 2009*) with a probability-based approach for repairing solutions, our method repairs only trial vectors $\mathbf{u}$ that satisfy two conditions: (i) they lost the tournament against their target vectors, but have a better objective function value, and (ii) they belong to a subproblem ($\mathbf{y}$) where no solution has been repaired in the current generation. These conditions aim to promote the repair of trial vectors with a higher potential to improve OF, exploring each subproblem independently without repairing similar solutions multiple times.

The repair method is explained in Algorithm 1. A mixed trial vector $\mathbf{u} = [\mathbf{x}_k^u, \mathbf{y}^u]^{\mathrm{T}}$ is defined, where only the $\mathbf{x}_k^u$ component is updated during the iterative repair process. As a result, the constraint violation degree vector and the gradient matrix can be found by

Eqs. (12) and (13), respectively. Both elements are updated in lines 3 and 4 of Algorithm 1. After removing all zero elements of $\mathbf{V}(\mathbf{x}_k^u, \mathbf{y}^u)$ and their corresponding values in $\nabla\mathbf{V}(\mathbf{x}_k^u, \mathbf{y}^u)$, the pseudoinverse $\nabla\mathbf{V}(\mathbf{x}_k^u, \mathbf{y}^u)^+$ is determined in line 6. Finally, the trial vector is updated between lines 7 and 9. The algorithm stops when all elements of vector $\mathbf{V}(\mathbf{x}_k^u, \mathbf{y}^u)$ become zero, indicating a feasible trial vector. Additionally, two stopping criteria are employed to prevent long repair cycles that could cause arbitrary increases in execution time. The first is the maximum number of iterations, which stops the process after $k_{max}$ iterations. The second is the minimum tolerance, which stops the process if the maximum variation in the trial vector during repair is lower than $T_{min}$.

$$\mathbf{V}(\mathbf{u}) = \mathbf{V}(\mathbf{x}_k^u, \mathbf{y}^u) \tag{12}$$

$$\nabla\mathbf{V}(\mathbf{u}) = \frac{(\mathbf{x}_k^u, \mathbf{y}^u)}{\partial\mathbf{x}_k^u}. \tag{13}$$

The repairing procedure is illustrated with the following example: consider a scenario with an inequality and an equality constraint, given by Eq. (14).

$$\begin{aligned} g_1(\mathbf{x}, y) &= x_1{}^2 + x_2{}^2 + y^2 - 12 \leq 0 \\ h_1(\mathbf{x}, y) &= x_1 + x_2 + y - 5.5 = 0. \end{aligned} \tag{14}$$

Assume $\mathbf{u} = [2\ 1\ 1]^T$ and an equality tolerance of $\varepsilon = 1 \times 10^{-04}$. In the first iteration (where $k = 1$), $\mathbf{x}_1^u = [2\ 1]^T$ and $\mathbf{y}^u = 1$. Therefore, $\mathbf{V}(\mathbf{x}_1^u, \mathbf{y}^u)$ and $\nabla\mathbf{V}(\mathbf{x}_1^u, \mathbf{y}^u)$ are computed by Eqs. (15) and (16), respectively.

$$\mathbf{V}(\mathbf{x}_1^u, \mathbf{y}^u) = \begin{bmatrix} \max(-6, 0) \\ -\max(1.5 - \varepsilon, 0) \end{bmatrix} = \begin{bmatrix} 0 \\ -1.4999 \end{bmatrix} \tag{15}$$

$$\nabla\mathbf{V}(\mathbf{x}_1^u, \mathbf{y}^u) = \begin{bmatrix} \frac{\partial g_1(\mathbf{x}_1^u, \mathbf{y}^u)}{\partial x_1} & \frac{\partial g_1(\mathbf{x}_1^u, \mathbf{y}^u)}{\partial x_2} \\ \frac{\partial h_1(\mathbf{x}_1^u, \mathbf{y}^u)}{\partial x_1} & \frac{\partial h_1(\mathbf{x}_1^u, \mathbf{y}^u)}{\partial x_2} \end{bmatrix} = \begin{bmatrix} 4 & 2 \\ 1 & 1 \end{bmatrix} \tag{16}$$

As can be seen, only $h_1(\mathbf{x}_1^u, \mathbf{y}^u)$ was violated. Therefore, the element of $g_1(\mathbf{x}_1^u, \mathbf{y}^u)$ in $\mathbf{V}(\mathbf{x}_1^u, \mathbf{y}^u)$ and their corresponding values in $\nabla\mathbf{V}(\mathbf{x}_1^u, \mathbf{y}^u)$ need to be removed. This leads to $\mathbf{V}(\mathbf{x}_1^u, \mathbf{y}^u) = -1.4999$. Then, $\nabla\mathbf{V}(\mathbf{x}_1^u, \mathbf{y}^u)$ and its pseudoinverse $\nabla\mathbf{V}(\mathbf{x}_1^u, \mathbf{y}^u)^+$ are computed as shown in Eq. (17).

$$\nabla\mathbf{V}(\mathbf{x}_1^u, \mathbf{y}^u) = [\,1 \quad 1\,] \Rightarrow \nabla\mathbf{V}(\mathbf{x}_1^u, \mathbf{y}^u)^+ = \begin{bmatrix} 0.5 \\ 0.5 \end{bmatrix} \tag{17}$$

Using Eq. (11), the vector $\mathbf{x}_2^u$ is generated as follows:

$$\mathbf{x}_2^u = \begin{bmatrix} 2 \\ 1 \end{bmatrix} - \begin{bmatrix} 0.5 \\ 0.5 \end{bmatrix}[-1.4999] = \begin{bmatrix} 2.75 \\ 1.75 \end{bmatrix}$$

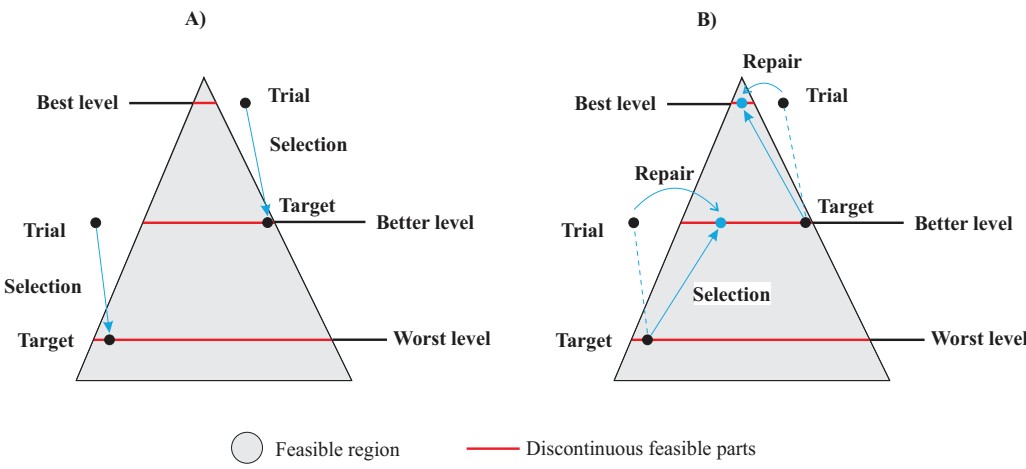

**Figure 2 Gradient-based repair method in MINLP context: (A) Standard DE variant with target-trial selection. (B) DE with gradient-based repair method.**

The updated vector $\mathbf{V}(\mathbf{x}_2^u, \mathbf{y}^u)$ is obtained by Eq. (18). As can be seen, the values of $(\mathbf{x}_2^u, \mathbf{y}^u)$ satisfy all constraints. Therefore, the trial vector has been successfully repaired, and its new values are $\mathbf{u} = [2.75\ 1.75\ 1]^T$.

$$\mathbf{V}(\mathbf{x}_2^u, \mathbf{y}^u) = \begin{bmatrix} \max(-0.3750, 0) \\ \max(0 - \varepsilon, 0) \end{bmatrix} = \begin{bmatrix} 0 \\ 0 \end{bmatrix} \tag{18}$$

Figure 2 provides an explanation of the gradient-based repair method within the context of DE, for a hypothetical MINLP problem. The feasible region is represented by a gray area, and includes discontinuous feasible parts generated by discrete variables shown as red lines. The objective space is defined by three contour curves: the worst level, a better level, and the best level. In Fig. 2A, the standard DE variant illustrates two instances of target-trial selection. Notably, the target solutions generated infeasible trials, making the target vectors the survivors in both cases, as indicated by the arrow directions. On the other hand, the gradient-based repair DE variant takes into account that both trial solutions have a better value in the objective space than their respective target vectors. This implies that both trial vectors must be repaired and compete again with their targets. Figure 2B illustrates how the repaired vectors enter the feasible regions and outperform their targets. This analysis highlights the advantages of the gradient-based repair method over the traditional selection method for finding feasible solutions within potentially promising subproblems, thereby promoting exploration within better feasible regions. Given the insensitivity of DE to constraints, repair might be essential to achieve feasibility in highly constrained subproblems.

## Overall implementation

Algorithm 2 describes G-DEmi. To generate the initial population $\mathbf{P}_g$ in step 1, random real and integer values for the solution vector $[\mathbf{x}, \mathbf{y}]$ are taken. The objective function $f(\mathbf{x}_g)$

| **Algorithm 2** G-DEmi framework. |
|---|

**Input:** $NP, CR, F, k_{max}, T_{min}, Eval_{max}$;

**Output:** best solution;

1: initialize population $\mathbf{P}_g$;

2: evaluate $f(\mathbf{x}_g)$ and $G(\mathbf{x}_g)$ for each individual in $\mathbf{P}_g$;

3: Eval $\leftarrow 1$;

4: $g \leftarrow 1$;

5: **while** Eval $<$ Eval$_{max}$ **do**

6:      $\mathbf{Y} \leftarrow \varnothing$;

7:      **for** each individual $\mathbf{x}_{g,i}$ in $\mathbf{P}_g$ **do**

8:         generate a trial vector $\mathbf{u}_{g,i}$;

9:         round the integer variables in $\mathbf{u}_{g,i}$;

10:         evaluate $f(\mathbf{u}_{g,i})$ and $G(\mathbf{u}_{g,i})$

11:         **if** $\mathbf{u}_{g,i}$ is better than $\mathbf{x}_{g,i}$ **then**

12:           store $\mathbf{u}_{g,i}$ into $\mathbf{P}_{g+1}$;

13:         **else if** $f(\mathbf{u}_{g,i}) < f(\mathbf{x}_{g,i})$ **and** $\mathbf{y}_{g,i}^{\mathrm{u}} \notin \mathbf{Y}$ **then**

14:           repair $\mathbf{u}_{g,i}$ according to Algorithm 1;

15:           evaluate $f(\mathbf{u}_{g,i})$ and $G(\mathbf{u}_{g,i})$

16:           **if** $\mathbf{u}_{g,i}$ is better than $\mathbf{x}_{g,i}$ **then**

17:             store $\mathbf{u}_{g,i}$ into $\mathbf{P}_{g+1}$;

18:           **end if**

19:           $\mathbf{Y} = \mathbf{Y} \cup \mathbf{y}_{g,i}^{\mathrm{u}}$;

20:         **end if**

21:         update Eval;

22:      **end for**

23:      $g \leftarrow g + 1$;

24: **end while**

and the constraint violation degree $G(\mathbf{x}_g)$ are then evaluated. In step 8, a trial vector $\mathbf{u}_{g,i}$ is generated for each target vector $\mathbf{x}_{g,i}$, using mutation and binomial crossover (rand/1/bin). The integer variables in $\mathbf{u}_{g,i}$ are rounded in step 9 before evaluating the vector in step 10. In the selection operation (steps 11 to 20), the trial vector is compared to its corresponding target vector. In step 11, the best solution is selected according to the feasibility rules described in "Base Methods". If the trial vector fails to improve its target but still has a lower objective function value ($f(\mathbf{u}_{g,i}) < f(\mathbf{x}_{g,i})$), and no other vector of the same subproblem has been repaired in the current generation ($\mathbf{y}_{g,i}^{\mathrm{u}} \notin \mathbf{Y}$), this trial vector is repaired in step 14. In step 16, following the feasibility rules, the better solution between the repaired vector and its target vector passed to the population of the next generation $\mathbf{P}_{g+1}$. Through these steps, G-DEmi generates a new population in each generation.

| Table 1 Main features of the 28 benchmark MINLP problems. | | | | | | | |
|---|---|---|---|---|---|---|---|
| Problem | $n$ | $n_1$ | $n_2$ | IC | EC | AC | OF type |
| F1 | 2 | 1 | 1 | 1 | 0 | 1 | Nonlinear |
| F2 | 3 | 1 | 2 | 1 | 0 | 1 | Nonlinear |
| F3 | 2 | 1 | 1 | 3 | 0 | 1 | Linear |
| F4 | 2 | 1 | 1 | 2 | 0 | 1 | Linear |
| F5 | 2 | 1 | 1 | 0 | 1 | 1 | Nonlinear |
| F6 | 2 | 1 | 1 | 2 | 0 | 1 | Nonlinear |
| F7 | 5 | 3 | 2 | 0 | 3 | 3 | Nonlinear |
| F8 | 8 | 5 | 3 | 6 | 0 | 6 | Linear |
| F9 | 8 | 5 | 3 | 6 | 0 | 6 | Linear |
| F10 | 8 | 5 | 3 | 6 | 0 | 6 | Linear |
| F11 | 15 | 12 | 3 | 5 | 0 | 4 | Nonlinear |
| F12 | 15 | 10 | 5 | 5 | 0 | 4 | Nonlinear |
| F13 | 6 | 4 | 2 | 0 | 4 | 4 | Nonlinear |
| F14 | 6 | 4 | 2 | 0 | 4 | 4 | Nonlinear |
| F15 | 10 | 7 | 3 | 8 | 0 | 4 | Nonlinear |
| F16 | 10 | 5 | 5 | 8 | 0 | 4 | Nonlinear |
| F17 | 6 | 3 | 3 | 6 | 0 | 3 | Nonlinear |
| F18 | 20 | 10 | 10 | 1 | 0 | 1 | Linear |
| F19 | 6 | 4 | 2 | 6 | 3 | 6 | Linear |
| F20 | 3 | 1 | 2 | 2 | 0 | 2 | Nonlinear |
| F21 | 5 | 3 | 2 | 6 | 0 | 2 | Nonlinear |
| F22 | 13 | 3 | 10 | 9 | 0 | 6 | Nonlinear |
| F23 | 7 | 3 | 4 | 9 | 0 | 4 | Nonlinear |
| F24 | 7 | 4 | 3 | 4 | 0 | 0 | Nonlinear |
| F25 | 4 | 2 | 2 | 3 | 0 | 2 | Nonlinear |
| F26 | 8 | 4 | 4 | 0 | 4 | 4 | Nonlinear |
| F27 | 8 | 3 | 5 | 0 | 3 | 3 | Nonlinear |
| F28 | 8 | 4 | 4 | 5 | 4 | 5 | Nonlinear |

# EXPERIMENTAL STUDY

## Settings

Twenty-eight benchmark MINLP minimization problems (F1-F28) were employed to thoroughly evaluate the performance of G-DEmi. They were proposed in *Liu et al. (2021a)*, *Datta & Figueira (2013)*, *Liu et al. (2023)* and *MINLPLib (2023)*. Some problems have several discontinuous feasible parts with different sizes, and the best solution is located in the smallest part (F1-F4). Other problems have equality constraints (F5, F7, F13, and F14), large dimensionality (F8-F12 and F15, F16, and F18), or binary variables (F17, F22, and F23). Finally, F19 and F26-F28 are highly constrained problems. The details of these test functions can be found in Supplemental File S1, and a summary of their main features is presented in Table 1, where $n$ is the number of variables of the problem; $n_1$ and $n_2$ are the

numbers of continuous and discrete variables, respectively; IC and EC are the numbers of inequality and equality constraints, respectively; and AC is the number of active constraints for the best-known solution so far.

A performance evaluation was conducted to compare G-DEmi with six state-of-the-art algorithms. Four different versions of DE were considered, including MI-EDDE (*Mohamed et al., 2019*), BOToP (*Liu et al., 2021b*), FROFI (*Wang et al., 2015*), and SADE (*Qin & Suganthan, 2005*; *Huang, Qin & Suganthan, 2006*), as well as two EAs based on different paradigms: PSOmv (*Wang, Zhang & Zhou, 2021*) based on particle swarm optimization, and EDAmv (*Molina-Pérez et al., 2022*) based on estimation of distribution. The versions of FROFI and SADE were combined with the cutting and repulsion strategy (CaR) proposed in *Liu et al. (2021a)*, resulting in FROFI-CaR and SADE-CaR, respectively. Furthermore, we conducted experiments to evaluate the effectiveness of the gradient-based repair method in other state-of-the-art DE variants. Two recent approaches were considered for this purpose: chaotic local search-based differential evolution (CJADE) (*Gao et al., 2019*) and DE with CaR plus surviving solutions (DE-CaR+S) (*Molina-Pérez et al., 2024*). CJADE has demonstrated high efficacy in optimization problems with continuous variables, while DE-CaR+S has reported remarkable results in MINLP problems. We performed the following comparative analysis: CJADE-CaR (CJADE with CaR strategy) *vs*. G-CJADEmi (CJADE with gradient-based repair) and DE-CaR+S *vs*. G-DEmi+S (DE with gradient-based repair plus surviving solutions).

Due to the stochastic nature of EAs, more than simply assessing the quality of outcomes is required to evaluate their performance. Hence, in this experiment, descriptive and inferential statistics were employed. To achieve this, each problem was solved with a maximum of 200,000 OF evaluations and 30 independent runs. The equality-constraint tolerance was set at $1 \times 10^{-04}$. A run was considered successful if $|f(\mathbf{x}_{best}) - f(\mathbf{x}^*)| \leq 1 \times 10^{-04}$, where $\mathbf{x}^*$ denotes the best-known solution and $\mathbf{x}_{best}$ represents the best solution generated by the algorithm. All algorithms and their instances were executed using the same computing resources: an Intel Core i7 -4790 CPU @3.6 GHz and 32 GB RAM with Windows 10 Enterprise N and MATLAB 2023a (The MathWorks, Natick, MA, USA).

The iRace parameter tuning tool was used to select parameter sets for all algorithms (*López-Ibáñez et al., 2016*). The tuning process used a representative set of test problems consisting of F1, F3, F5, F7, F9, F10, F12, F14, F16, F18, F20, F22, F24, F26, and F28. For each algorithm, 2,000 experiments were conducted in iRace. The parameters used in each algorithm are presented in Table 2, where population size is denoted by NP, crossover rate by CR, scaling factor by $F$, maximum number of repair iterations by $k_{\max}$, minimum repair tolerance by $T_{\min}$, proportion of population partitions by $P_a$, and initial tolerance for equality constraints by $e_0$. Parameters $t_c$ and $c_p$ are the controllers of the e-constrained method. For algorithms with a learning process, $L$ denotes the learning iteration count, $\alpha$ is the learning rate, $r$ represents the chaotic search radius, $c$ is the rate of parameter adaptation, $p$ is the greediness of the mutation strategy, $\varepsilon$ is the link parameter, $r_M$ is the mutation rate, $\beta$ represents the mutation factor, $w$ denotes the number of histogram bins, $\varepsilon$

**Table 2 Algorithm parameters.**

| Algorithm | NP | CR | F or β | $k_{max}$ | $T_{min}$ | T | Additional parameters |
|---|---|---|---|---|---|---|---|
| MI-EDDE | 60 | 0.8086 | [0.6268–0.9459] | – | – | – | $P_a = 10$, $e_0 = 4.3137$ |
| BOToP | 40 | {0.1, 0.2, 1.0} | {0.6, 0.8, 1.0} | – | – | – | – |
| SADE-CaR | 60 | Adaptative | Adaptative | – | – | 800 | – |
| FROFI-CaR | 60 | {0.1, 0.2, 1.0} | {0.6, 0.8, 1.0} | – | – | 400 | – |
| G-DEmi | 50 | 0.793 | 0.7 | 50 | $1 \times 10^{-64}$ | – | – |
| CJADE-CaR | 50 | Adaptative | Adaptative | – | – | 400 | $r = 0.0851$, $c = 0.3686$, $p = 0.05$, $L = 50$, $\rho = 0.9733$. |
| G-CJADEmi | 50 | Adaptative | Adaptative | 50 | $1 \times 10^{-64}$ | – | $r = 0.1447$, $c = 0.0304$, $p = 0.1$, $L = 100$, $\rho = 0.9188$. |
| DE-CaR+S | 50 | [0.5154–0.9154] | [0.4806–0.5501] | – | – | 400 | $\delta = 0.4 \times 10^{-04}$, $NR = 5$, $l_1 = 1$, $l_2 = 3$. |
| G-DEmi+S | 60 | [0.7241–0.9241] | [0.5295–0.6726] | 80 | $1 \times 10^{-64}$ | – | $NR = 3$, $l_1 = 1$, $l_2 = 3$. |
| PSOmv | 300 | – | – | – | – | – | $A_c = 1.2482$, $\alpha = 0.0173$. |
| EDAmv | 100 | – | [0.2165–0.6306] | – | – | – | $w = 4$, $\varepsilon_b = 2.4722$, $t_c = 1000$, $c_p = 7.2427$, $\varepsilon_p = 0.3508$, $r_M = 0.6603$. |

represents the end bins parameter, $A_c$ is the acceleration coefficient, and $\rho$ represents the shrinking parameter. The failure threshold is denoted by $T$. In replacement strategies, NR indicates the number of surviving replacements, and $\delta$ is the improvement tolerance. Additionally, $l_1$ and $l_2$ represent the count of weighted difference vectors.

As described above, G-DEmi rounds the values of integer variables after the rand/1/bin operation. The value of $F$ suggested by iRace (higher than the 0.5 recommended by *Storn & Price, 1997*) can be effective in reducing the repeated integer values after rounding and encouraging the exploration of different discontinuous feasible parts. The value of CR indicates that crossover is also significant in this scheme to prevent accelerated convergence. The population of 50 individuals coincides with several successful DE experiments for MINLP problems (*Mohamed et al., 2019*; *Liu et al., 2021a*).

## Benchmark results and analysis

The experimental results were analyzed using descriptive statistics. The selected indicators are widely recognized in the field for assessing algorithms, providing a comprehensive view of their performance (*Liu et al., 2021b*, *2021a*). The feasible rate (FR) represents the percentage of runs in which the algorithm finds at least one feasible solution, whereas the successful rate (SR) represents the percentage of runs in which the algorithm finds the optimal solution within the given tolerance. Ave and Std denote the average and the standard deviation of the best OF values of each problem from its 30 independent runs. If an algorithm cannot achieve 100% FR, then the Ave and Std values are denoted as "NA". Mean FR and mean SR denote the mean feasible rate and the mean successful rate achieved by the algorithm over the twenty-eight test problems. T represents the average execution time for each problem in seconds.

Inferential statistics were employed to evaluate the statistical significance of the differences in results. A comparison between G-DEmi and the other algorithms was carried out using the Wilcoxon Rank-Sum Test (WRST) with a 0.05 significance level

**Table 3 Performance comparison conducted over 30 independent runs with 200,000 OF evaluations for MI-EDDE, BOToP, SADE-CaR, FROFI-CaR, and G-DEmi (Part 1 of 2).**

| Prob. | Parameter | MI-EDDE | BOToP | SADE-CaR | FROFI-CaR | G-DEmi |
|---|---|---|---|---|---|---|
| F1 | FR | **100** | **100** | **100** | **100** | **100** |
| | SR | 0 | **100** | **100** | **100** | **100** |
| | Ave | 1.70E+01 + | 1.30E+01 | 1.30E+01 ≈ | 1.30E+01 ≈ | 1.30E+01 |
| | Std | 4.62E−3 | 7.23E−15 | 9.57E−13 | 3.03E−11 | 7.23E−15 |
| | T | 6.20E+00 | 6.06E+00 | 5.30E+00 | 7.28E+00 | 3.60E+00 |
| F2 | FR | **100** | **100** | **100** | **100** | **100** |
| | SR | 0 | **100** | **100** | **100** | **100** |
| | Ave | 2.00E+00 + | 1.00E+00 ≈ | 1.00E+00 ≈ | 1.00E+00 ≈ | 1.00E+00 |
| | Std | 0.00E+00 | 0.00E+00 | 0.00E+00 | 0.00E+00 | 0.00E+00 |
| | T | 6.38E+00 | 3.35E+00 | 1.07E+01 | 6.23E+00 | 3.67E+00 |
| F3 | FR | **100** | **100** | **100** | **100** | **100** |
| | SR | 16.67 | 96.67 | **100** | **100** | **100** |
| | Ave | −3.58E+00 + | −3.98E+00 ≈ | −4.00E+00 ≈ | −4.00E+00 ≈ | −4.00E+00 |
| | Std | 1.90E−01 | 9.13E−02 | 0.00E+00 | 0.00E+00 | 0.00E+00 |
| | T | 6.26E+00 | 7.33E+00 | 5.39E+00 | 7.60E+00 | 3.72E+00 |
| F4 | FR | **100** | **100** | **100** | **100** | **100** |
| | SR | **100** | **100** | **100** | **100** | **100** |
| | Ave | −6.00E+00 ≈ | −6.00E+00 ≈ | −6.00E+00 ≈ | −6.00E+00 ≈ | −6.00E+00 |
| | Std | 0.00E+00 | 0.00E+00 | 0.00E+00 | 0.00E+00 | 0.00E+00 |
| | T | 6.33E+00 | 4.92E+00 | 5.30E+00 | 7.50E+00 | 3.43E+00 |
| F5 | FR | **100** | **100** | **100** | **100** | **100** |
| | SR | **100** | 93.33 | **100** | **100** | **100** |
| | Ave | 2.50E−01 ≈ | 3.16E−01 ≈ | 2.50E−01 ≈ | 2.50E−01 ≈ | 2.50E−01 |
| | Std | 8.47E−17 | 2.51E−01 | 8.47E−17 | 8.47E−17 | 2.54E−05 |
| | T | 6.42E+00 | 2.39E+00 | 5.62E+00 | 4.87E+00 | 3.48E+00 |
| F6 | FR | **100** | **100** | **100** | **100** | **100** |
| | SR | 96.67 | **100** | **100** | **100** | **100** |
| | Ave | −6.71E+03 ≈ | −6.78E+03 ≈ | −6.78E+03 ≈ | −6.78E+03 ≈ | −6.78E+03 |
| | Std | 2.66E+02 | 2.78E−12 | 2.78E−12 | 2.78E−12 | 2.78E−12 |
| | T | 6.48E+00 | 5.74E+00 | 5.25E+00 | 5.82E+00 | 3.60E+00 |
| F7 | FR | **100** | 83.33 | **100** | **100** | **100** |
| | SR | 0 | 16.67 | 0 | **100** | **100** |
| | Ave | 1.00E+00 + | NA | 1.00E+00 + | 2.11E−01 ≈ | 2.11E−01 |
| | Std | 0.00E+00 | NA | 0.00E+00 | 1.46E−16 | 1.13E−16 |
| | T | 6.41E+00 | 1.90E+00 | 1.07E+01 | 7.79E+00 | 7.45E+00 |
| F8 | FR | **100** | **100** | **100** | **100** | **100** |
| | SR | 0 | 0 | 90 | 86.67 | **96.67** |
| | Ave | 7.21E+03 + | 7.09E+03 + | 7.06E+03 ≈ | 7.06E+03 ≈ | 7.06E+03 |
| | Std | 1.60E+02 | 3.90E+01 | 9.33E+00 | 1.03E+01 | 6.90E+00 |
| | T | 6.78E+00 | 8.77E+00 | 9.67E+00 | 7.75E+00 | 1.52E+01 |
| F9 | FR | **100** | **100** | **100** | **100** | **100** |

(Continued)

| Table 3 (continued) | | | | | | |
|---|---|---|---|---|---|---|
| Prob. | Parameter | MI-EDDE | BOToP | SADE-CaR | FROFI-CaR | G-DEmi |
| | SR | 0 | 50.00 | 86.67 | 73.33 | **100** |
| | Ave | 7.81E+03 + | 7.15E+03 + | 7.09E+03 ≈ | 7.13E+03 + | 7.08E+03 |
| | Std | 4.03E+02 | 1.21E+02 | 1.72E+01 | 8.67E+01 | 0.00E+00 |
| | T | 6.47E+00 | 8.16E+00 | 9.77E+00 | 7.79E+00 | 8.72E+00 |
| F10 | FR | **100** | **100** | **100** | **100** | **100** |
| | SR | 0 | 53.33 | 96.67 | 83.33 | **100** |
| | Ave | 8.83E+03 + | 7.44E+03 + | 7.15E+03 ≈ | 7.24E+03 ≈ | 7.13E+03 |
| | Std | 1.29E+02 | 3.95E+02 | 8.52E+01 | 2.66E+02 | 0.00E+00 |
| | T | 6.59E+00 | 7.31E+00 | 9.72E+00 | 7.82E+00 | 8.12E+00 |
| F11 | FR | **100** | **100** | **100** | **100** | **100** |
| | SR | 0 | 26.67 | 6.67 | **93.33** | 60.00 |
| | Ave | 3.36E+01 + | 3.38E+01 ≈ | 3.40E+01 + | 3.35E+01 − | 3.35E+01 |
| | Std | 6.39E−02 | 5.60E−01 | 7.90E−01 | 3.31E−02 | 4.95E−02 |
| | T | 7.02E+00 | 1.30E+01 | 9.48E+00 | 1.27E+01 | 5.49E+01 |
| F12 | FR | **100** | **100** | **100** | **100** | **100** |
| | SR | 46.67 | 46.67 | 10.00 | 26.67 | **100** |
| | Ave | 5.88E+01 + | 4.22E+01 ≈ | 4.21E+01 + | 4.20E+01 + | 4.17E+01 |
| | Std | 2.65E+01 | 8.26E−01 | 1.83E−01 | 1.64E−01 | 3.69E−13 |
| | T | 6.88E+00 | 1.30E+01 | 9.53E+00 | 1.27E+01 | 4.57E+01 |
| F13 | FR | 80 | **100** | **100** | **100** | **100** |
| | SR | 0 | 0 | 0 | 60.00 | **100** |
| | Ave | NA | 8.95E+03 + | 8.94E+03 + | 8.89E+03 + | 8.88E+03 |
| | Std | NA | 2.13E+01 | 4.47E+01 | 5.06E+00 | 5.47E−12 |
| | T | 6.83E+00 | 4.07E+00 | 1.25E+01 | 7.80E+00 | 1.60E+01 |
| F14 | FR | 86.67 | 96.67 | **100** | **100** | **100** |
| | SR | 16.67 | 0 | 23.33 | 63.33 | **100** |
| | Ave | NA | NA | 8.95E+03 + | 8.95E+03 + | 8.95E+03 |
| | Std | NA | NA | 5.05E+00 | 2.57E+00 | 3.70E−12 |
| | T | 6.74E+00 | 2.82E+00 | 1.26E+01 | 7.82E+00 | 6.87E+00 |

**Note:**
Values in boldface indicate the best result.

(*Derrac et al., 2011*). Additionally, we applied the Bonferroni correction to control the family-wise error rate that arises from multiple pairwise comparisons (*Dinno, 2015*). The symbols indicating the superiority, inferiority, or similarity of our proposal compared to each competitor are denoted by "+," "−," and "≈", respectively.

Tables 3 and 4 present the statistical values of each problem, in separate rows. For example, in problem F1, all algorithms successfully found feasible solutions across all executions, yielding 100% FR. The success rates indicate that MI-EDDE failed every run (SR = 0), whereas the other competing algorithms consistently achieved the optimal solution (SR = 100). Based on WRST, G-DEmi was comparable to the other competitors,

**Table 4 Performance comparison conducted over 30 independent runs with 200,000 OF evaluations for MI-EDDE, BOToP, SADE-CaR, FROFI-CaR, and G-DEmi (Part 2 of 2).**

| Prob. | Parameter | MI-EDDE | BOToP | SADE-CaR | FROFI-CaR | G-DEmi |
|---|---|---|---|---|---|---|
| F15 | FR | **100** | **100** | **100** | **100** | **100** |
| | SR | 13.33 | 0 | 0 | 26.67 | **96.67** |
| | Ave | 2.97E+01 + | 3.05E+01 + | 2.88E+01 + | 2.87E+01 + | 2.84E+01 |
| | Std | 1.18E+00 | 9.50E-01 | 3.12E-08 | 1.86E−01 | 7.53E−02 |
| | T | 6.54E+00 | 6.12E+00 | 9.18E+00 | 8.03E+00 | 1.16E+01 |
| F16 | FR | **100** | **100** | **100** | **100** | **100** |
| | SR | 66.67 | 0 | 100 | 100 | 100 |
| | Ave | 2.94E+01 + | 3.11E+01 + | 2.85E+01 ≈ | 2.85E+01 ≈ | 2.85E+01 |
| | Std | 1.24E+00 | 3.79E−15 | 1.10E−10 | 1.87E−14 | 9.67E−15 |
| | T | 6.66E+00 | 6.33E+00 | 9.76E+00 | 8.02E+00 | 9.60E+00 |
| F17 | FR | **100** | 0 | **100** | **100** | **100** |
| | SR | 13.33 | 0 | 0 | 0 | 100 |
| | Ave | 6.06E+00 + | NA | 8.62E+00 + | 7.67E+00 + | 6.01E+00 |
| | Std | 4.13E−02 | NA | 3.28E+00 | 2.07E−14 | 2.71E−15 |
| | T | 6.41E+00 | 4.32E+00 | 1.06E+01 | 7.82E+00 | 5.04E+00 |
| F18 | FR | **100** | **100** | **100** | **100** | **100** |
| | SR | 33.33 | 86.67 | 76.67 | **100** | 100 |
| | Ave | −2.17E+01 + | −2.17E+01 + | −2.17E+01 + | −2.17E+01 ≈ | −2.17E+01 |
| | Std | 1.51E−02 | 1.42E−02 | 1.06E−02 | 6.09E−13 | 1.72E−08 |
| | T | 6.72E+00 | 9.85E+00 | 1.04E+01 | 8.82E+00 | 1.51E+01 |
| F19 | FR | 0 | 0 | 0 | 0 | **100** |
| | SR | 0 | 0 | 0 | 0 | 100 |
| | Ave | NA | NA | NA | NA | 9.92E+01 |
| | Std | NA | NA | NA | NA | 0.00E+00 |
| | T | 6.52E+00 | 6.99E+00 | 1.33E+01 | 7.20E+00 | 8.17E+00 |
| F20 | FR | **100** | 90 | **100** | **100** | **100** |
| | SR | 73.33 | 40 | 76.67 | **100** | 100 |
| | Ave | −5.54E+00 + | NA | −5.57E+00 + | −5.68E+00 ≈ | −5.68E+00 |
| | Std | 2.53E−01 | NA | 2.39E−01 | 9.03E−16 | 9.03E−16 |
| | T | 6.40E+00 | 1.21E+01 | 1.17E+01 | 7.75E+00 | 3.85E+00 |
| F21 | FR | **100** | **100** | **100** | **100** | **100** |
| | SR | **100** | **100** | 100 | 100 | 100 |
| | Ave | −3.07E+04 ≈ | −3.07E+04 ≈ | −3.07E+04 ≈ | −3.07E+04 ≈ | −3.07E+04 |
| | Std | 1.38E−11 | 1.48E−11 | 1.45E−11 | 1.48E−11 | 1.44E−11 |
| | T | 6.40E+00 | 1.18E+01 | 1.18E+01 | 7.91E+00 | 4.43E+00 |
| F22 | FR | **100** | **100** | **100** | **100** | **100** |
| | SR | 93.33 | **100** | 100 | 100 | 100 |
| | Ave | −1.50E+01 ≈ | −1.50E+01 ≈ | −1.50E+01 ≈ | −1.50E+01 ≈ | −1.50E+01 |
| | Std | 6.24E−02 | 0.00E+00 | 0.00E+00 | 0.00E+00 | 0.00E+00 |
| | T | 6.86E+00 | 1.59E+01 | 1.39E+01 | 9.02E+00 | 4.10E+00 |
| F23 | FR | **100** | **100** | **100** | **100** | **100** |

(Continued)

| Table 4 (continued) | | | | | | |
|---|---|---|---|---|---|---|
| Prob. | Parameter | MI-EDDE | BOToP | SADE-CaR | FROFI-CaR | G-DEmi |
| | SR | **100** | 50 | **100** | **100** | 93.33 |
| | Ave | 3.56E+00 ≈ | 3.58E+00 + | 3.56E+00 ≈ | 3.56E+00 ≈ | 3.56E+00 |
| | Std | 1.36E−15 | 3.87E−02 | 1.36E−15 | 1.36E−15 | 5.61E−03 |
| | T | 6.60E+00 | 5.88E+00 | 1.28E+01 | 7.92E+00 | 5.00E+00 |
| F24 | FR | **100** | **100** | **100** | **100** | **100** |
| | SR | 96.67 | 50 | **100** | **100** | **100** |
| | Ave | 6.83E+02 ≈ | 6.83E+02 + | 6.83E+02 ≈ | 6.83E+02 ≈ | 6.83E+02 |
| | Std | 2.92E−02 | 8.13E−02 | 7.00E−14 | 4.22E−14 | 6.68E−14 |
| | T | 6.70E+00 | 7.88E+00 | 1.16E+01 | 7.97E+00 | 6.22E+00 |
| F25 | FR | **100** | **100** | **100** | **100** | **100** |
| | SR | **100** | **100** | **100** | 13.33 | **100** |
| | Ave | 6.06E+03 ≈ | 6.06E+03 ≈ | 6.06E+03 ≈ | 6.06E+03 + | 6.06E+03 |
| | Std | 9.25E−13 | 9.25E−13 | 9.25E−13 | 5.71E+00 | 9.25E−13 |
| | T | 6.49E+00 | 9.84E+00 | 1.06E+01 | 7.77E+00 | 4.20E+00 |
| F26 | FR | 0 | 0 | 0 | 0 | **100** |
| | SR | 0 | 0 | 0 | 0 | **100** |
| | Ave | NA | NA | NA | NA | 1.39E−07 |
| | Std | NA | NA | NA | NA | 2.71E−07 |
| | T | 6.72E+00 | 1.15E+01 | 1.22E+01 | 8.03E+00 | 5.68E+00 |
| F27 | FR | 0 | 13.33 | 3.33 | 30 | **100** |
| | SR | 0 | 0 | 0 | 30 | **100** |
| | Ave | NA | NA | NA | NA | 5.98E+00 |
| | Std | NA | NA | NA | NA | 0.00E+00 |
| | T | 6.64E+00 | 2.74E+00 | 1.21E+01 | 7.79E+00 | 5.70E+00 |
| F28 | FR | 0 | 36.67 | 0 | **100** | **100** |
| | SR | 0 | 0 | 0 | **100** | **100** |
| | Ave | NA | NA | NA | 6.06E+00 ≈ | 6.06E+00 |
| | Std | NA | NA | NA | 2.71E−15 | 2.71E−15 |
| | T | 6.74E+00 | 2.19E+00 | 1.21E+01 | 1.07E+01 | 8.74E+00 |
| Mean FR | | 84.52 | 82.86 | 85.83 | 90.36 | **100.00** |
| Mean SR | | 38.10 | 46.79 | 59.52 | 73.45 | **98.10** |
| WRST [+/≈/−] | | [20/8/0] | [17/11/0] | [13/15/0] | [10/17/1] | |

**Note:**
Values in boldface indicate the best result.

except for MI-EDDE, which was significantly inferior. Additionally, MI-EDDE had the fastest average execution time of 5.91 s. The remaining problems were analyzed similarly.

Mean FR, mean SR, and the final count of WRST are provided in the last section of Table 4. It can be observed that G-DEmi reached feasible solutions in all executions for all problems, 100% of Mean FR, surpassing the 84.52% for MI-EDDE, 82.86% for BOToP, 85.83% for SADE-CaR, and 90.36% for FROFI-CaR. Note that the highly constrained condition of problems F19 and F26-F28 had no influence on the FR of G-DEmi. However,

this condition significantly affected the performance of the competing algorithms. In terms of mean SR, G-DEmi obtained the highest value at 98.10%, followed by FROFI-CaR at 73.45%, SADE-CaR at 59.52%, BOToP at 46.79%, and MI-EDDE at 38.10%. The final count of WRST shows that G-DEmi significantly outperformed MI-EDDE in 20 problems, whereas both algorithms had similar results in eight problems, with MI-EDDE never surpassing the performance of G-DEmi. BOToP is significantly outperformed by G-DEmi in 17 problems, equaled in 11 problems, and never achieving better results than G-DEmi. Similarly, FROFI-CaR was surpassed by G-DEmi in 10 problems, whereas its results were similar for 17 problems, and it only significantly exceeded G-DEmi in one problem (F11).

In this experiment, there were some outlier values that should be noted. For problems F1-F3, the solutions of MI-EDDE tended to converge towards local optima deliberately located in the most accessible discontinuous feasible parts. In contrast, problems F7 and F13, with equality constraints, affected the performance of other algorithms but not the G-DEmi proposal. In problems F8-F10, certain points were attractive to the algorithm population, with MI-EDDE and POToP showing the worst performances. Problem F11 has more than 10 real variables and a high sensitivity of the objective function to small variations in the decision variables. As a result, G-DEmi did not reach the solution before finishing the execution in 40% of the cases. Remarkable advantages of G-DEmi over PSOmv and EDAmv are shown in Tables S5 and S6 in Supplementary File S2. PSOmv and EDAmv obtained Mean FR values of 85.83% and 88.45%, and Mean SR values of 14.17% and 37.74%, respectively. G-DEmi outperformed PSOmv in 25 problems, with only three instances of similar performance. On the other hand, EDAmv was outperformed in 21 problems and only achieved similar performances in seven problems. G-DEmi did not exhibit inferior performance with respect to PSOmv and EDAmv in any case.

The experiments involving the gradient-based repair method with other DE variants are reported in Tables 5 and 6. These tables present the results of CJADE-CaR *vs*. G-CJADEmi and DE-CaR+S *vs*. G-DEmi+S. The comparison highlights that G-CJADEmi outperformed CJADE-CaR, achieving 100% of Mean FR and 90.48% of Mean SR compared to 98.81% and 69.05%, respectively. G-CJADEmi exhibited better results in 12 problems and similar results in 14 problems, but CJADE-CaR performed better in problems F15 and F23. In the second competition, G-DEmi+S surpassed DE-CaR+S with a Mean FR of 99.52% *vs*. 90.24% and a Mean SR of 91.55% *vs*. 83.93%. G-DEmi+S achieved better results in seven problems, with no significant difference in 19 problems, and was outperformed in F12 and F23 by DE-CaR+S. In general, the variants with gradient-based repair exhibited notable superiority, with G-DEmi as the most promising in terms of feasibility and success.

Additionally, an analysis was conducted to examine the convergence of G-DEmi during the evolutionary process. Three problems were selected for this: F3 with two variables, F12 with 15 variables and high sensitivity of OF to changes in decision variables, and F27 highly constrained. Figure 3 shows the convergence curves of the median solutions in terms of solution quality for problem F12. Each curve starts from the generation of at least one feasible solution. A star marker on each curve shows when the algorithm reached the

**Table 5  Performance comparison conducted over 30 independent runs with 200,000 OF evaluations for CJADE-CaR, G-CJADEmi, DE-CaR +S, and G-DEmi+S (Part 1 of 2).**

| Problem | Parameter | CJADE-CaR | G-CJADEmi | DE-CaR+S | G-DEmi+S |
|---|---|---|---|---|---|
| F1 | FR | **100** | **100** | **100** | **100** |
|  | SR | **100** | **100** | **100** | **100** |
|  | Ave | 1.30E+01 ≈ | 1.30E+01 | 1.30E+01 ≈ | 1.30E+01 |
|  | Std | 5.27E−12 | 7.23E−15 | 6.79E−12 | 7.23E−15 |
|  | T | 5.21E+00 | 1.37E+01 | 1.16E+01 | 4.87E+00 |
| F2 | FR | **100** | **100** | **100** | **100** |
|  | SR | **100** | **100** | **100** | **100** |
|  | Ave | 1.00E+00 ≈ | 1.00E+00 | 1.00E+00 ≈ | 1.00E+00 |
|  | Std | 0.00E+00 | 0.00E+00 | 0.00E+00 | 0.00E+00 |
|  | T | 6.20E+00 | 1.09E+01 | 1.49E+01 | 1.03E+01 |
| F3 | FR | **100** | **100** | **100** | **100** |
|  | SR | **100** | **100** | **100** | **100** |
|  | Ave | −4.00E+00 ≈ | −4.00E+00 | −4.00E+00 ≈ | −4.00E+00 |
|  | Std | 0.00E+00 | 0.00E+00 | 0.00E+00 | 0.00E+00 |
|  | T | 5.76E+00 | 1.35E+01 | 1.52E+01 | 1.10E+01 |
| F4 | FR | **100** | **100** | **100** | **100** |
|  | SR | **100** | **100** | **100** | **100** |
|  | Ave | −6.00E+00 ≈ | −6.00E+00 | −6.00E+00 ≈ | −6.00E+00 |
|  | Std | 0.00E+00 | 0.00E+00 | 0.00E+00 | 0.00E+00 |
|  | T | 5.77E+00 | 1.24E+01 | 1.69E+01 | 1.03E+01 |
| F5 | FR | **100** | **100** | **100** | **100** |
|  | SR | **100** | **100** | **100** | **100** |
|  | Ave | 2.50E−01 ≈ | 2.50E−01 | 2.50E−01 ≈ | 2.50E−01 |
|  | Std | 8.47E−17 | 8.47E−17 | 8.47E−17 | 8.47E−17 |
|  | T | 5.69E+00 | 1.36E+01 | 1.47E+01 | 1.03E+01 |
| F6 | FR | **100** | **100** | **100** | **100** |
|  | SR | **100** | **100** | **100** | **100** |
|  | Ave | −6.78E+03 ≈ | −6.78E+03 | −6.78E+03 ≈ | −6.78E+03 |
|  | Std | 2.78E−12 | 2.78E−12 | 2.78E−12 | 2.78E−12 |
|  | T | 5.85E+00 | 1.32E+01 | 1.66E+01 | 1.06E+01 |
| F7 | FR | **100** | **100** | **100** | **100** |
|  | SR | 0 | **93.33** | 90 | **100** |
|  | Ave | 9.59E−01 + | 2.43E−01 | 2.83E−01 ≈ | 2.11E−01 |
|  | Std | 3.26E−02 | 1.36E−01 | 2.20E−01 | 1.13E−16 |
|  | T | 5.56E+00 | 2.04E+01 | 1.60E+01 | 1.86E+01 |
| F8 | FR | **100** | **100** | **100** | **100** |
|  | SR | 83.33 | **96.67** | 96.67 | **100** |
|  | Ave | 7.06E+03 + | 7.06E+03 | 7.06E+03 + | 7.06E+03 |
|  | Std | 1.15E+01 | 9.33E+00 | 9.33E+00 | 2.78E−12 |
|  | T | 3.74E+00 | 4.10E+01 | 2.28E+01 | 3.23E+01 |
| F9 | FR | **100** | **100** | **100** | **100** |

(Continued)

| Problem | Parameter | CJADE-CaR | G-CJADEmi | DE-CaR+S | G-DEmi+S |
|---|---|---|---|---|---|
| | SR | 50 | **100** | **100** | **100** |
| | Ave | 7.12E+03 + | 7.08E+03 | 7.08E+03 ≈ | 7.08E+03 |
| | Std | 4.73E+01 | 0.00E+00 | 5.13E−12 | 0.00E+00 |
| | T | 4.19E+00 | 2.84E+01 | 2.06E+01 | 2.51E+01 |
| F10 | FR | **100** | **100** | **100** | **100** |
| | SR | 73.33 | **100** | 96.67 | **100** |
| | Ave | 7.18E+03 + | 7.13E+03 | 7.13E+03 + | 7.13E+03 |
| | Std | 1.42E+02 | 0.00E+00 | 8.05E−04 | 0.00E+00 |
| | T | 4.47E+00 | 2.77E+01 | 1.98E+01 | 2.54E+01 |
| F11 | FR | **100** | **100** | **100** | **100** |
| | SR | 20 | **63.33** | 100 | 53.33 |
| | Ave | 3.49E+01 + | 3.36E+01 | 3.35E+01 ≈ | 3.36E+01 |
| | Std | 1.41E+00 | 1.15E−01 | 5.52E−06 | 6.63E−02 |
| | T | 3.56E+00 | 3.44E+01 | 2.12E+01 | 7.06E+01 |
| F12 | FR | **100** | **100** | **100** | **100** |
| | SR | 23.33 | **43.33** | 100 | 76.67 |
| | Ave | 4.22E+01 + | 4.37E+01 | 4.17E+01 − | 4.18E+01 |
| | Std | 3.33E−01 | 9.92E+00 | 5.77E−08 | 1.42E−01 |
| | T | 3.66E+00 | 2.48E+01 | 1.85E+01 | 5.64E+01 |
| F13 | FR | **100** | **100** | **100** | **100** |
| | SR | 53.33 | **100** | **100** | **100** |
| | Ave | 8.89E+03 + | 8.88E+03 | 8.88E+03 ≈ | 8.88E+03 |
| | Std | 1.36E+01 | 5.55E−12 | 5.55E−12 | 5.55E−12 |
| | T | 5.15E+00 | 2.93E+01 | 1.90E+01 | 8.08E+01 |
| F14 | FR | **100** | **100** | **100** | **100** |
| | SR | 90 | **100** | **100** | **100** |
| | Ave | 8.95E+03 ≈ | 8.95E+03 | 8.95E+03 ≈ | 8.95E+03 |
| | Std | 1.33E+00 | 3.70E−12 | 3.70E−12 | 3.70E−12 |
| | T | 5.43E+00 | 3.03E+01 | 1.78E+01 | 4.65E+01 |

**Note:**
Values in boldface indicate the best result.

**Table 6 Performance comparison conducted over 30 independent runs with 200,000 OF evaluations for CJADE-CaR, G-CJADEmi, DE-CaR +S, and G-DEmi+S (Part 2 of 2).**

| Problem | Parameter | CJADE-CaR | G-CJADEmi | DE-CaR+S | G-DEmi+S |
|---|---|---|---|---|---|
| F15 | FR | **100** | **100** | **100** | **100** |
| | SR | **90** | 16.67 | **73.33** | 16.67 |
| | Ave | 2.84E+01 − | 2.87E+01 | 2.85E+01 ≈ | 2.87E+01 |
| | Std | 1.05E−01 | 1.56E−01 | 1.86E−01 | 1.56E−01 |
| | T | 4.58E+00 | 2.03E+01 | 1.81E+01 | 3.95E+01 |
| F16 | FR | **100** | **100** | **100** | **100** |
| | SR | **100** | **100** | **100** | **100** |

(Continued)

| Table 6 (continued) | | | | | |
|---|---|---|---|---|---|
| Problem | Parameter | CJADE-CaR | G-CJADEmi | DE-CaR+S | G-DEmi+S |
| | Ave | 2.85E+01 ≈ | 2.85E+01 | 2.85E+01 ≈ | 2.85E+01 |
| | Std | 1.03E−14 | 1.43E−14 | 1.56E−12 | 1.36E−14 |
| | T | 5.38E+00 | 1.80E+01 | 1.85E+01 | 3.37E+01 |
| F17 | FR | **100** | **100** | **100** | **100** |
| | SR | 26.67 | **100** | **100** | **100** |
| | Ave | 8.21E+00 + | 6.01E+00 | 6.01E+00 ≈ | 6.01E+00 |
| | Std | 2.96E+00 | 2.71E−15 | 3.50E-09 | 2.71E−15 |
| | T | 3.78E+00 | 1.43E+01 | 1.42E+01 | 1.59E+01 |
| F18 | FR | **100** | **100** | **100** | **100** |
| | SR | 46.67 | **90** | **100** | **100** |
| | Ave | −2.17E+01 ≈ | −2.17E+01 | −2.17E+01 ≈ | −2.17E+01 |
| | Std | 1.67E−03 | 2.43E−02 | 4.34E−09 | 9.24E−15 |
| | T | 4.53E+00 | 1.76E+01 | 3.75E+01 | 5.12E+01 |
| F19 | FR | 66.67 | **100** | 10 | **100** |
| | SR | 6.67 | **93.33** | 6.67 | **90** |
| | Ave | NA | 9.98E+01 | NA | 1.00E+02 |
| | Std | NA | 2.06E+00 | NA | 2.48E+00 |
| | T | 3.86E+00 | 1.49E+01 | 1.47E+01 | 1.64E+01 |
| F20 | FR | **100** | **100** | **100** | **100** |
| | SR | **100** | **100** | **100** | 96.67 |
| | Ave | −5.68E+00 ≈ | −5.68E+00 | −5.68E+00 ≈ | −5.67E+00 |
| | Std | 9.03E−16 | 9.03E−16 | 9.03E−16 | 1.08E−01 |
| | T | 6.25E+00 | 1.48E+01 | 1.67E+01 | 1.05E+01 |
| F21 | FR | **100** | **100** | **100** | **100** |
| | SR | **100** | **100** | **100** | **100** |
| | Ave | −3.07E+04 ≈ | −3.07E+04 | −3.07E+04 ≈ | −3.07E+04 |
| | Std | 1.34E−11 | 1.48E−11 | 1.48E−11 | 1.48E−11 |
| | T | 5.89E+00 | 1.48E+01 | 1.73E+01 | 1.35E+01 |
| F22 | FR | **100** | **100** | **100** | **100** |
| | SR | **100** | **100** | **100** | **100** |
| | Ave | −1.50E+01 ≈ | −1.50E+01 | −1.50E+01 ≈ | −1.50E+01 |
| | Std | 0.00E+00 | 0.00E+00 | 0.00E+00 | 0.00E+00 |
| | T | 6.46E+00 | 1.49E+01 | 1.97E+01 | 1.49E+01 |
| F23 | FR | **100** | **100** | **100** | **100** |
| | SR | **90** | 46.67 | **100** | 80 |
| | Ave | 3.56E+00 − | 3.69E+00 | 3.56E+00 − | 3.56E+00 |
| | Std | 6.75E−03 | 3.19E-01 | 1.36E−15 | 9.00E−03 |
| | T | 5.88E+00 | 1.36E+01 | 1.65E+01 | 1.52E+01 |
| F24 | FR | **100** | **100** | **100** | **100** |
| | SR | **100** | **100** | **100** | **100** |
| | Ave | 6.83E+02 ≈ | 6.83E+02 | 6.83E+02 ≈ | 6.83E+02 |

(Continued)

| Problem | Parameter | CJADE-CaR | G-CJADEmi | DE-CaR+S | G-DEmi+S |
|---------|-----------|-----------|-----------|----------|----------|
| | Std | 9.44E−14 | 2.99E−14 | 1.10E−13 | 1.32E−13 |
| | T | 6.09E+00 | 1.43E+01 | 1.71E+01 | 1.50E+01 |
| F25 | FR | **100** | **100** | **100** | **100** |
| | SR | **100** | **100** | 80 | **100** |
| | Ave | 6.06E+03 ≈ | 6.06E+03 | 6.06E+03 + | 6.06E+03 |
| | Std | 9.25E−13 | 9.25E−13 | 4.21E+00 | 9.25E−13 |
| | T | 6.01E+00 | 1.67E+01 | 1.95E+01 | 1.24E+01 |
| F26 | FR | **100** | **100** | 0 | **90** |
| | SR | 80 | **100** | 0 | **90** |
| | Ave | 1.12E−02 + | 3.36E−08 | NA | NA |
| | Std | 4.23E−02 | 1.42E−07 | NA | NA |
| | T | 4.74E+00 | 1.61E+01 | 5.06E+01 | 1.34E+01 |
| F27 | FR | **100** | **100** | 56.67 | **96.67** |
| | SR | 0 | **90** | 6.67 | **66.67** |
| | Ave | 6.95E+00 + | 5.99E+00 | NA | NA |
| | Std | 2.42E−01 | 4.63E−02 | NA | NA |
| | T | 5.39E+00 | 1.58E+01 | 4.36E+01 | 1.56E+01 |
| F28 | FR | **100** | **100** | 60 | **100** |
| | SR | 0 | **100** | 0 | **93.33** |
| | Ave | 1.66E+03 + | 6.06E+00 | NA | 6.21E+00 |
| | Std | 1.29E+03 | 2.71E−15 | NA | 6.04E−01 |
| | T | 5.25E+00 | 4.43E+01 | 4.14E+01 | 1.85E+01 |
| Mean FR | | 98.81 | **100** | 90.24 | **99.52** |
| Mean SR | | 69.05 | **90.48** | 83.93 | **91.55** |
| WRST [+/≈/−] | | | | | |
| | | [12/14/2] | | [7/19/2] | |

**Note:**
Values in boldface indicate the best result.

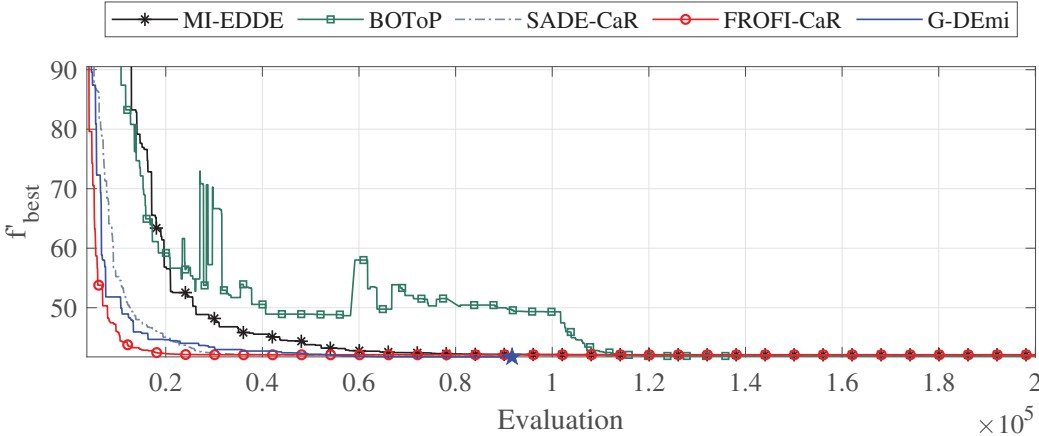

**Figure 3 Convergence curves of the median solutions from the 30 runs for problem F12.**

global solution. It is observed that G-DEmi converged rapidly and found the global optimum before 100,000 evaluations (blue point). In contrast, the other algorithms consumed 200,000 evaluations without achieving this result. In Supplemental File S2, Figs. S1 and S2 show the curves for F3 and F27, indicating an even more favorable outcome for G-DEmi.

Likewise, the impact of the parameters $k_{max}$ and $T_{min}$ on the performance of G-DEmi was analyzed through the described problems. For each problem, 30 executions of 200,000 evaluations were carried out using different combinations of them, while the other parameters remained fixed at previously reported values. In Supplemental File S2, Tables S2, S3, and S4 show the average execution times [s] and the average OF values for each instance. The best values are highlighted in bold. In general, the performance was adversely affected by the option of no repair ($k_{max} = 0$), and better performances were seen as $k_{max}$ increased and $T_{min}$ decreased. However, it is important to note that a longer repair process resulted in higher time consumption. Therefore, the execution times could be a determining factor for reasonable values such as those used. Values of $k_{max}$ equal to 50 and 100, and $T_{min}$ of $1 \times 10^{-60}$ and $1 \times 10^{-80}$ proved effective in this study. The consistency of the performance suggests that both parameters have a robust behavior concerning the algorithm's performance.

In summary, the comprehensive analysis of the applied metrics reveals that the general performance of G-DEmi surpassed the other state-of-the-art algorithms. Notably, WRST confirmed the statistical significance of the observed differences in results. The variety of the solved problems indicates that the gradient-based method contributes to the performance of G-DEmi in a wide range of MINLP problems. Even in highly constrained problems with challenging feasible regions, the proposed approach has consistently demonstrated remarkable success rates. The successful integration of the gradient-based repair method across both standard and advanced versions of DE emphasizes the robustness and effectiveness of this approach within the context of this evolutionary algorithm.

It is important to note that, unlike the methods that promote diversity using additional variation operators such as composite trial generation, self-adaptive parameters, or dynamic parameters, G-DEmi uses the conventional variation operators of DE/rand/1/bin. The success of G-DEmi is due to the efficient search process within each subproblem provided by the gradient-based repair method. The proposed repair method compensates, to a large extent, the insensibility of DE to constraints in the context of MINLP problems.

## REAL-WORLD CASE STUDY

### Description

An important number of real-world optimization problems are MINLP problems. For this reason, we selected a well-known real problem as a case study to evaluate the performance of G-DEmi. The subway optimization problem (SOP) was proposed in *Bock & Longman (1982)* for the New York subway system as a dynamic nonlinear mixed-integer control problem. The aim is to minimize the energy consumption of a train between two stations,

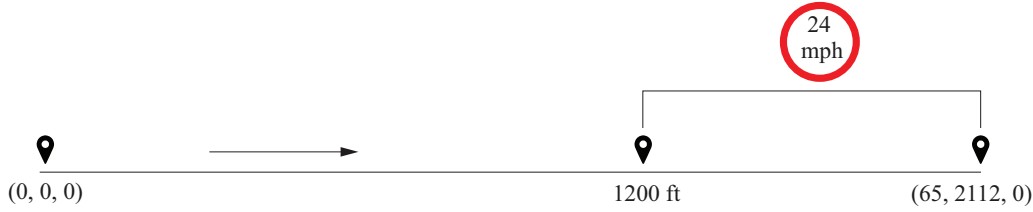

**Figure 4 Diagram of the SOP with velocity constraint.**

taking into account the dynamic model and other constraints. The decision variables for this problem include the operating modes for the route (integer variables) as well as the switching times for each operating mode (real variables). A significant challenge in solving this problem is the presence of mixed variables, specifically in the form of time-dependent integer variables (*Belotti et al., 2013*).

In this study, we address the variant of SOP illustrated in Fig. 4 and described in *Sager (2005)*, *Sager, Bock & Reinelt (2009)*, *Lee & Leyffer (2011)*. The subway starts with zero values of time, position, and velocity, and is required to come to a complete stop (zero velocity) at a position 2,112 ft ahead, within a maximum time of 65 s. A path constraint is added to a subset of the track, requiring that the subway's velocity never exceeds 24 mph (35.2 ft/s) from 1,200 ft to the end of the track. The initial conditions of the problem are represented by the values $(0, 0, 0)$ for time, position, and velocity, while the final conditions correspond to $(65, 2{,}112, 0)$.

## Optimal control problem

The SOP is expressed as an optimal control problem in Eq. (19):

$$
\min_{\mathbf{x},w} \quad \int_0^{t_f} L(\mathbf{x}, w)\, dt
$$
$$
\begin{aligned}
subject\ to:\ & \dot{x}_0 = x_1 \\
& \dot{x}_1 = f_1(\mathbf{x}, w) \\
& x_1 \le 35.2 \text{ if } x_0 \ge 1{,}200 \\
& \mathbf{x}(0) = (0,0)^T,\ \mathbf{x}(t_f) = (2{,}112,0)^T \\
& w(t) \in \{1, 2, 3, 4\},\ t \in [0, t_f].
\end{aligned}
\tag{19}
$$

The arrival time of the subway to the next station is denoted by the terminal time $t_f = 65$ s. The differential states $x_0(\cdot)$ and $x_1(\cdot)$ represent the position of the train in ft and velocity in ft/s, respectively. The subway can operate under four modes: $w(\cdot) = 1$ in serial, $w(\cdot) = 2$ in parallel, $w(\cdot) = 3$ in coasting, and $w(\cdot) = 4$ in braking. The Lagrange term for each of the four modes is defined in Eqs. (20)–(22):

$$
L(\mathbf{x}, 1) =
\begin{cases}
\dfrac{ep_1}{3{,}600}, & \text{if } x_1 \le v_1 \\[2mm]
\dfrac{ep_2}{3{,}600}, & \text{if } v_1 < x_1 \le v_2 \\[2mm]
\dfrac{e \sum_{i=0}^{5} c_i(1)(0.1\gamma x_1)^{-i}}{3{,}600}, & \text{if } v_2 < x_1
\end{cases}
\tag{20}
$$

$$L(\mathbf{x}, 2) = \begin{cases} 0, & \text{if } x_1 \leq v_2 \\ \frac{ep_3}{3,600}, & \text{if } v_2 < x_1 \leq v_3 \\ \frac{e \sum_{i=0}^{5} c_i(2)(0.1\gamma x_1 - 1)^{-i}}{3,600}, & \text{if } v_3 < x_1 \end{cases} \tag{21}$$

$$L(\mathbf{x}, 3) = 0, L(\mathbf{x}, 4) = 0. \tag{22}$$

The acceleration function $f(\mathbf{x}, w)$ for each operating mode is represented by Eqs. (23)–(26):

$$f_1(\mathbf{x}, 1) = \begin{cases} \frac{gea_1}{w_{eff}}, & \text{if } x_1 \leq v_1 \\ \frac{gea_2}{w_{eff}}, & \text{if } v_1 < x_1 \leq v_2 \\ \frac{g(eT(x_1, 1) - R(x_1))}{w_{eff}}, & \text{if } v_2 < x_1 l \end{cases} \tag{23}$$

$$f_1(\mathbf{x}, 2) = \begin{cases} 0, & \text{if } x_1 \leq v_2 \\ \frac{gea_3}{w_{eff}}, & \text{if } v_2 < x_1 \leq v_3 \\ \frac{g(eT(x_1, 2) - R(x_1))}{w_{eff}}, & \text{if } v_3 < x_1 \end{cases} \tag{24}$$

$$f_1(\mathbf{x}, 3) = -\frac{gR(x_1)}{w_{eff}} - C \tag{25}$$

$$f_1(\mathbf{x}, 4) = -u_{max}. \tag{26}$$

In Eqs. (27)–(29), the drag force per car ($R$) and the tractive force per car ($T$) are denoted in lb. The specific parameter values used in this model can be found in Supplemental File S3.

$$R(x_1) = ca\gamma^2 x_1^2 + \frac{bW}{2,000}\gamma x_1 + \frac{1.3}{2,000}W + 116 \tag{27}$$

$$T(x_1, 1) = \sum_{i=0}^{5} b_i(1)(0.1\gamma x_1 - 0.3)^{-i} \tag{28}$$

$$T(x_1, 2) = \sum_{i=0}^{5} b_i(2)(0.1\gamma x_1 - 1)^{-i}. \tag{29}$$

### Decision variables

The present problem comprises eight operational states along the subway track. Each state is defined by a time interval in which one of the four operating modes is active. These states can be mathematically expressed by Eq. (30).

$$w(t) = \begin{cases} w_0 & \text{for } t_0 \leq t < t_1 \\ w_1 & \text{for } t_1 \leq t < t_2 \\ w_2 & \text{for } t_2 \leq t < t_3 \\ \vdots & \qquad \vdots \\ w_7 & \text{for } t_7 \leq t < t_f. \end{cases} \tag{30}$$

Since it is already known that $t_0 = 0$, $t_f = 65$, and $w_0 = 1$ (serial mode) and $w_7 = 4$ (braking mode), the decision variable vector is defined as
$\mathbf{d} = [t_1, t_2, t_3, t_4, t_5, t_6, t_7, w_1, w_2, w_3, w_4, w_5, w_6].$

**Table 7 SOP results in 30 independent runs.**

| Parameter | FR | Best | Median | Worst | Ave | Std Desv | Ave. Time per run [min] |
|---|---|---|---|---|---|---|---|
| G-DEmi | 100 | 1.369E+00 | 1.406E+00 | **1.459E+00** | 1.409E+00 | **1.937E−02** | 5.76E+01 |
| G-CJADEmi | **100** | **1.355E+00** | **1.388E+00** | 1.462E+00 | **1.397E+00** | 2.656E−02 | 9.03E+01 |
| G-DEmi+S | 100 | 1.376E+00 | 1.414E+00 | 1.489E+00 | 1.420E+00 | 3.056E−02 | 3.73E+01 |
| MI-EDDE | 0 | – | – | – | – | – | 1.29E+01 |
| SADE-CaR | 0 | – | – | – | – | – | 2.73E+01 |
| FROFI-CaR | 0 | – | – | – | – | – | 5.04E+00 |
| CJADE-CaR | 0 | – | – | – | – | – | 3.01E+00 |
| DE-CaR+S | 0 | – | – | – | – | – | 2.85E+00 |

Note:
Values in boldface indicate the best result.

### Numeric scheme

In this work, the single shooting method is used to convert the boundary value problem into an initial value problem. Therefore, the final conditions $\mathbf{x}(t_f)$ are considered as two additional equality constraints. The dynamic model is then solved using Runge Kutta 45 with a relative error tolerance of $1 \times 10^{-04}$. The path constraint is activated at an unknown time, 1,200 ft away from the initial position, and an event detection technique is used to determine the exact time of constraint activation. The bisection method is employed with a tolerance of $1 \times 10^{-06}$ to enclose the solution between the times before and after the event. Similarly, the same process is used to determine the exact time when the subway stops after the braking mode, and when the finishing line is crossed.

### Results and analysis

The case study was solved using MI-EDDE, SADE-CaR, FROFI-CaR, CJADE-CaR, G-CJADEmi, DE-CaR+S, G-DEmi+S, and G-DEmi algorithms. BOToP was unsuitable for this problem because its procedure involves evaluating solutions without integrity conditions, which was impossible in this case. Each algorithm was executed 30 times with 100,000 evaluations per run. Table 7 shows the final results of each competing algorithm. As can be seen, MI-EDDE, SADE-CaR, CJADE-CaR, DE-CaR+S, and FROFI-CaR failed to find feasible solutions in any of the runs. In contrast, all DE variants with gradient-based repair consistently produced feasible solutions in all 30 runs, with standard deviation values consistently below 0.05. In this study, G-CJADEmi reported the best results, yielding 1.355 kWh as the best fitness solution, with a corresponding decision vector of:

$$\mathbf{d}^* = [0,\ 2.6747071,\ 15.9101142,\ 32.4524515,\ 34.1250172,\ 37.3175333,\ 49.6483266,$$
$$57.8272705, 65,\ 1,\ 2,\ 1,\ 4,\ 3,\ 1,\ 3,\ 4].$$

In *Sager (2005)*, the best solution reported was 1.384 kWh. However, it is important to note that a fair comparison with Sager's result is challenging due to the unknown parameters of the numerical methods employed in that particular previous work.

Figure 5 shows the time-velocity curves of the best solutions obtained in this experiment, including the solution reported in *Sager (2005)*. The current operation mode is

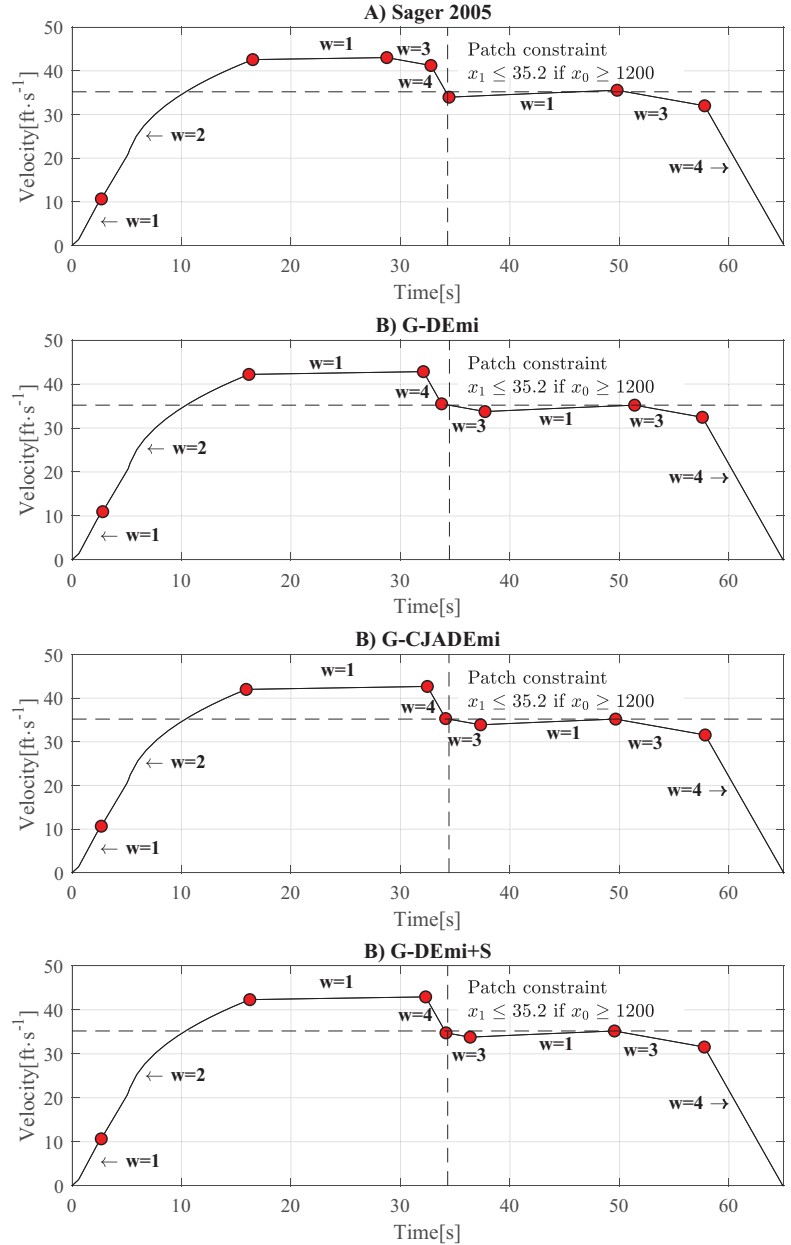

**Figure 5 Time-velocity curves of the best solution from: (A)** *Sager (2005)*. **(B) G-DEmi. (C) G-CJADEmi. (D) G-DEmi+CaR. The vertical dashed line indicates $x_0 = 1200$ ft, and the horizontal dashed line corresponds to $x_1 = 32.5$ ft/s.** The upper right quadrant of the axes depicts the path constraint.

represented by $w$, and the red points indicate operation mode changes. As can be seen, the behavior of the best solutions is based on three key factors that contribute to reducing energy consumption: (i) returning to cruising speed ($w = 1$) after the initial acceleration (flattened zone of the curve), (ii) maintaining the path constraint active, and (iii) entering a coasting zone ($w = 3$) before the final braking operation. The behavior exhibited by all of these solutions primarily consisted of these factors.

The previous results verified the effectiveness of DE variants with gradient-based repair in solving a real-world optimization problem, such as SOP. However, as in several test problems, these variants had a higher running time. This suggests that the gradient-based repair method could be a highly time-consuming process, particularly when dealing with dynamic models as constraints. In Supplementary File S4, a time complexity analysis was conducted for the G-DEmi proposal, demonstrating how the repair process can significantly increase the algorithm's complexity. Therefore, other strategies, such as surrogate or coevolutionary methods, can be combined with the repair method to reduce the required time.

## CONCLUSIONS

In the MINLP problems, several discontinuous feasible parts are generated, which can be analyzed as individual subproblems. The number of generated subproblems can be considerable in many cases. Consequently, it becomes imperative to guide the search towards promising subproblems, as well as an efficient exploration and exploitation within each subproblem. However, traditional evolutionary algorithms exhibit insensitivity to constraints, severely impacting their effectiveness in exploring highly constrained subproblems, constituting a search bias towards subproblems with more accessible feasible regions.

To address this issue, we proposed a variant of DE called G-DEmi, which incorporates a gradient-based repair method for MINLP problems. The repair method aims to fix infeasible solutions within different subproblems by using the gradient information of the constraint set.

We conducted extensive experiments to evaluate the performance of G-DEmi. This included testing it on 28 benchmark problems with diverse features, as well as applying it to a real-world MINLP problem. The results consistently demonstrated that G-DEmi outperformed several state-of-the-art algorithms. Notably, G-DEmi achieved high success rates even in highly constrained problems where other algorithms failed to find feasible solutions. These findings indicate that the gradient-based repair method significantly contributes to the overall performance of G-DEmi. Furthermore, the gradient-based repair method was successfully applied in other DE variants, highlighting its robustness and effectiveness in a more general context of this evolutionary algorithm.

Another important point is that G-DEmi does not require additional strategies in the variation operators to promote diversity. Instead, it successfully uses the conventional variation operators of DE/rand/1/bin. The key to its success lies in the efficient search process within each subproblem, which is due to the gradient-based repair method. Therefore, this approach significantly mitigates the insensitivity of DE to constraints in the context of MINLP problems.

The repair method could be highly time-consuming, particularly in real-world problems with complex constraints. Therefore, its combination with strategies such as surrogate or coevolutionary methods is recommended for future work to reduce the time required for the repair process. However, gradient-based repair methods become impractical when the constraint set lacks proper behavior or differentiation. In those cases,

exploring alternative approaches, such as direct methods for constraint repairing, is advisable. The integration of the gradient-based repair method into other evolutionary approaches could be a significant contribution, allowing for the assessment of its capabilities in a broader context. Additionally, future research will encompass the evaluation of G-DEmi in scalability, large-scale, and multi-objective mixed-integer problems.

### Funding

This work was supported by the Mexican National Council of Science and Technology (CONACyT), the Secretariat of Research and Postgraduate Studies of the National Polytechnic, through projects Nos. 20232444 and 20231601. The funders had no role in study design, data collection and analysis, decision to publish, or preparation of the manuscript.

### Grant Disclosures

The following grant information was disclosed by the authors:
Mexican National Council of Science and Technology: CONACyT.
National Polytechnic: Nos. 20232444 and 20231601.

### Competing Interests

The authors declare that they have no competing interests.

### Author Contributions

- Daniel Molina-Pérez conceived and designed the experiments, performed the experiments, performed the computation work, prepared figures and/or tables, authored or reviewed drafts of the article, and approved the final draft.
- Edgar Alfredo Portilla-Flores analyzed the data, prepared figures and/or tables, authored or reviewed drafts of the article, and approved the final draft.
- Efrén Mezura-Montes conceived and designed the experiments, analyzed the data, authored or reviewed drafts of the article, and approved the final draft.
- Eduardo Vega-Alvarado performed the experiments, performed the computation work, authored or reviewed drafts of the article, and approved the final draft.
- María Bárbara Calva-Yañez analyzed the data, authored or reviewed drafts of the article, and approved the final draft.

### Data Availability

The raw data and code are available in the Supplemental Files.

### Supplemental Information

Supplemental information for this article can be found online at http://dx.doi.org/10.7717/peerj-cs.2095#supplemental-information.

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
