# Peer review of "Efficiently handling constraints in mixed-integer nonlinear programming problems using gradient-based repair differential evolution"

_PeerJ Computer Science, doi:10.7717/peerj-cs.2095_

## Round 0.1 · original submission · Major Revisions

There are issues which, as a result, lead to major revision of the present manuscript. The issues involve the addition of newer references, a justification of the parameter setting that you use, and a more detailed experimental analysis which include also a smaller number of evaluations and some analysis with other state-of-the-art methods like from referee two. (Other than she/he, I would suggest that you implement two or more methods and that you choose the most promising ones.) Other issues you see in the comments of the referees.

Reviewer 1 ·

Basic reporting

This paper proposed a variant of differential evolution that modifies infeasible solutions by using the gradient information of the constraint set. The previous studies were well researched and summarized. The experiments were conducted comprehensively. I have some suggestions for improvement.

1. In Figure 1, it is unclear what the curve represents and can be misunderstood. If you draw contour lines of the evaluation values, the same integer variable area should have the same evaluation value.

2. The use of bold and non-bold indices is inconsistent and needs to be unified. (e.g., Equation 9, 10, and 11)

Experimental design

3. There is a lack of discussion regarding the hyperparameters of the proposed method. In particular, it should be discussed how the trade-off between the accuracy of the repair and the number of evaluations required changes depending on the setting of $k_{max}$ and $T_{min}$ in the gradient-based repair method.

4. In Section 4.1, the maximum number of evaluations of 200,000 may be too high. Increasing the number of evaluations may bias the results in favor of the proposed method because more evaluations can be used in the gradient-based repair method. From the current results, it is difficult to determine whether the proposed method is effective when the maximum number of evaluations is small. It is desirable to either add low-budget results or display the transition of evaluation values as shown in Figure 7 of [1].

[1] Yuanhao Liu, Zan Yang, Danyang Xu, Haobo Qiu, Liang Gao, A surrogate-assisted differential evolution for expensive constrained optimization problems involving mixed-integer variables, Information Sciences, Volume 622, 2023, Pages 282-302, ISSN 0020-0255.

Validity of the findings

No comment

Additional comments

No comment

Reviewer 2 ·

Basic reporting

See below.

Experimental design

See below.

Validity of the findings

See below.

Additional comments

The paper introduces Mixed Integer Nonlinear Programming (MINLP) problems, emphasizing their complex nature due to the combination of continuous and discrete variables alongside nonlinear functions. These problems often contain various discontinuous feasible parts, leading to challenges for Evolutionary Algorithms (EAs), which typically generate infeasible solutions. To address this, the paper proposes a variant of Differential Evolution called G-DEmi, which utilizes a Gradient-based repair method to handle infeasible solutions within subproblems. Extensive experiments demonstrated that G-DEmi outperformed various state-of-the-art algorithms across a diverse set of MINLP problems, showcasing its efficacy in managing constraints and discontinuous feasible parts. The method's ability to effectively explore within each subproblem without needing additional diversity promotion strategies further solidifies its efficiency for handling MINLP scenarios.
However, this paper has some comments needed to be rigorously addressed as listed as follows:
1. The references require an update with the removal of outdated literature and the addition of recent work from top-tier journals. Care should be taken to rectify any missing or incorrect information, such as page numbers, while ensuring uniform adherence to journal standards. Additionally, the related work section should incorporate information on algorithms utilizing gradient-based repair methods and common MINLP problems.
2. A graphical representation elucidating the key concept of the paper should be included to enhance understanding.
3. The experimental setup, encompassing the utilized equipment and other pertinent details, must be clearly elucidated. It remains ambiguous whether the performance comparison was conducted fairly. Were all compared algorithms executed on the same PC?
4. A comprehensive performance comparison with previously proposed state-of-the-art methods from other literature is conspicuously absent. The comparison should encompass algorithms proposed for the test problems, state-of-the-art algorithms employing the gradient-based repair method, and the latest DE variants introduced in the last three years. Some of the important ones include Multiple Elite Individual Guided Piecewise Search-Based Differential Evolution; Bi-objective Elite Differential Evolution Algorithm for Multivalued Logic Networks; Chaotic Local Search-Based Differential Evolution Algorithms for Optimization; Multiobjective Optimized Deployment of Edge-Enabled Wireless Visual Sensor Networks for Target Coverage; and Biobjective Task Scheduling for Distributed Green Data Centers.
5. G-DEmi employs specific parameter settings of NP = 50, CR = 0.793, and F = 0.7. The rationale behind the adoption of these parameter configurations should be clarified by the authors.
6. It is advisable for the authors to present the parameter settings of G-Demi and the comparison algorithms in a tabular format for easy reference and comparison.
7. Tables 2, 3, and 4 necessitate the consistent use of scientific notation for data representation, ensuring precision and clarity in the presentation of results.

---

## Round 0.2 · Major Revisions

You can see that the referees are now more positive about the manuscript, but they ask for some more comments on your side. Therefore, I put this paper at major revisions.

Reviewer 2 has requested that you cite specific references. You may add them if you believe they are especially relevant. However, I do not expect you to include these citations, and if you do not include them, this will not influence my decision.

Reviewer 1 ·

Basic reporting

Thank you for revising your manuscript in response to the feedback. You have addressed some of the reviewers' suggestions, but there is still one issue that needs to be addressed as follows.

The contour lines of the objective function defined by continuous and integer variables are usually shown as [1, Figure 3]. On the other hand, in Figure 1 of the authors' paper, the contour lines are not discrete on the vertical (integer) axis, which falsely implies that the evaluation values vary smoothly. This problem also occurs in Figure 2 and needs to be corrected.

[1] Tea Tušar, Dimo Brockhoff, and Nikolaus Hansen, “Mixed-integer benchmark problems for single-and bi-objective optimization,” In Proceedings of the Genetic and Evolutionary Computation Conference (GECCO '19).

Experimental design

no comment

Validity of the findings

no comment

Additional comments

no comment

Reviewer 2 ·

Basic reporting

This paper proposes a variant of differential evolution with a gradient-based repair method for solving mixed integer nonlinear programming (MINLP) problems, which involve both continuous and discrete decision variables and nonlinear functions.
I notice that this article has undergone a revision, resulting in improved readability and enriched content. The proposed enhancements to DE result in significant improvements, particularly in nonlinear problem domains. However, there are still some issues that need to be addressed. Some comments are listed as follows:
1. It is hoped to keep in lowercase except for proper nouns. For instance, words like “Differential Evolution” “Gradient-based”.
2. It is recommended to maintain the present simple tense throughout the article for consistency.
3. By the end of introduction, the motivation is not highlighted, and how to validate this optimization framework is not explicitly stated.
4. Punctuation in all mathematical formulas needs to be checked.
5. Recent references are missing about new soft computing strategies and swarm intelligence, such as DOI: 10.1007/s44196-021-00030-z; DOI: 10.1038/s41598-023-40080-1; DOI: 1109/TSMC.2019.2956121; DOI: 10.1109/TNNLS.2021.3105901. These works should be included when reviewing the related work.

6. In the pseudocode of Algorithm 1, the stopping criteria of three choices should be clarified.
7. In experiment section, the table format is expected to be improved, and explanations for the occurrence of some outlier values would be appreciated.
8. It is suggested to augment the computational complexity analysis of the algorithm in conjunction with time consumption comparisons.
9. In terms of writing, the document exhibits some light typos and hard-to-read sentences. A thorough revision is recommended to eliminate grammar errors or misprints and enhance overall readability. Some lengthy sentences are quite perplexing; it is advisable to break down such overly long sentences into shorter ones.

Experimental design

See above.

Validity of the findings

See above.

Additional comments

See above.

---

## Round 0.3 · Minor Revisions

There are still minor issues with the paper. You are advised to do a final check of the whole paper.

Reviewer 2 ·

Basic reporting

The quality of the paper has significantly improved. However, it still contains several minor formatting errors that need to be addressed. For example, the term "otherwise" in line 214 should not be in italicized typeface, and the subfigure numbering in Figure 2 is inconsistent, alternating between "(A)" and "A)". The authors are advised to conduct a thorough review of the entire text to identify and correct any similar issues.

Experimental design

See above.

Validity of the findings

See above.

Additional comments

See above.

---

## Round 0.4 · accepted · Accept

I confirm that the authors have addressed all of the reviewers' comments and that the paper can go to publication. Only one thing should be solved, the citation to the irace paper should be Manuel L\'opez-Ib\'a\~nez, Leslie P\'erez C\'aceres, J\'er\'emie Dubois--Lacoste, Mauro Birattari and Thomas St{\"u}tzle.
The irace Package: Iterated Race for Automatic Algorithm
Configuration. Operations Research Perspectives, 3:43--58, 2016.